# A selective inhibitor of mitofusin 1-βIIPKC association improves heart failure outcome in rats

Julio C.B. Ferreira [1,2], Juliane C. Campos[1], Nir Qvit[2], Xin Qi [2,3], Luiz H.M. Bozi[1], Luiz R.G. Bechara[1], Vanessa M. Lima[1], Bruno B. Queliconi [4], Marie-Helene Disatnik[2], Paulo M.M. Dourado[5], Alicia J. Kowaltowski[4] & Daria Mochly-Rosen [2]

We previously demonstrated that beta II protein kinase C (βIIPKC) activity is elevated in failing hearts and contributes to this pathology. Here we report that βIIPKC accumulates on the mitochondrial outer membrane and phosphorylates mitofusin 1 (Mfn1) at serine 86. Mfn1 phosphorylation results in partial loss of its GTPase activity and in a buildup of fragmented and dysfunctional mitochondria in heart failure. βIIPKC siRNA or a βIIPKC inhibitor mitigates mitochondrial fragmentation and cell death. We confirm that Mfn1-βIIPKC interaction alone is critical in inhibiting mitochondrial function and cardiac myocyte viability using SAMβA, a rationally-designed peptide that selectively antagonizes Mfn1-βIIPKC association. SAMβA treatment protects cultured neonatal and adult cardiac myocytes, but not Mfn1 knockout cells, from stress-induced death. Importantly, SAMβA treatment re-establishes mitochondrial morphology and function and improves cardiac contractility in rats with heart failure, suggesting that SAMβA may be a potential treatment for patients with heart failure.

[1] Department of Anatomy, Institute of Biomedical Sciences, University of Sao Paulo, Sao Paulo 05508-000 SP, Brazil. [2] Department of Chemical and Systems Biology, Stanford University School of Medicine, Stanford 94305-5174 CA, USA. [3] Department of Physiology & Biophysics, Case Western Reserve University, Cleveland 44106 OH, USA. [4] Departamento de Bioquímica, Instituto de Química, Universidade de Sao Paulo, Sao Paulo 05508-000 SP, Brazil. [5] Heart Institute, University of Sao Paulo, Sao Paulo 05403-010 SP, Brazil. Correspondence and requests for materials should be addressed to J.C.B.F. (email: jcesarbf@usp.br) or to D.M.-R. (email: mochly@stanford.edu)

Despite advances in clinical and pharmacological interventions, acute myocardial infarction with subsequent left ventricular dysfunction and heart failure continues to be a major cause of morbidity and mortality worldwide[1,2]. Therefore, the identification of novel therapeutic targets that improve cardiac function in patients with myocardial infarction-induced heart failure remains a major priority.

Protein kinases are key players in cellular signaling[3]. They work as intracellular nodes where signals converge to and serve as multi-effector triggers and/or brakes. Through phosphorylation of specific substrates, protein kinases can regulate a wide array of intracellular pathways that control cardiac metabolism, contractility, remodeling, and survival. Therefore, they are attractive molecular targets for cardiovascular diseases. However, the vast majority of protein kinase inhibitors targets its ATP binding pocket, a highly conserved region across the kinome, and often induces cardiotoxicity by inhibiting unintended kinases[4,5]. Moreover, regardless of its upstream signaling, the same protein kinase can activate multiple signaling pathways simultaneously (essential and detrimental) during disease progression and therefore affect the effectiveness and safety of PKC inhibitors in the long-term. The screen or design of molecules that competitively disrupt a specific protein–protein interaction (kinase-substrate) has been considered an important approach to develop more feasible drugs that selectively affect only detrimental kinase-substrate interactions in cardiac pathophysiology[5–7].

Protein kinase C (PKC) is a family of closely related serine-threonine protein kinases involved in a variety of acute and chronic cardiovascular diseases (i.e., ischemia-reperfusion injury, hypertension, and heart failure)[8]. We have previously demonstrated that treatment of isolated hearts, rodents, or humans with rational design peptides that inhibit protein–protein interaction between a specific PKC and its anchor protein protects the heart against acute ischemic injuries[9,10]. We have also demonstrated that activation of beta II PKC (βIIPKC), but not other PKCs, contributes to heart failure pathophysiology in rodents[11,12]. Likewise, βIIPKC activity is elevated in human failing hearts[11]. More recently, we provided evidence that a rationally designed peptide against βIIPKC, βII$_{V5-3}$, improves cardiac function in rats with hypertension-induced or post-myocardial infarction-induced heart failure[12,13]. This peptide inhibits all βIIPKC activities (referred to as a global βIIPKC inhibitor in this paper).

Here, we use the global βIIPKC inhibitor to identify mitofusin 1 (Mfn1) as a downstream βIIPKC substrate involved in heart failure progression. Mfn1 is a ubiquitous and well-conserved GTPase responsible for regulating mitochondrial dynamics and bioenergetics. We show that βIIPKC associates with Mfn1. Using a rationally designed peptide that selectively antagonizes Mfn1-βIIPKC association (SAMβA), we determine the contribution of Mfn1-βIIPKC interaction and the resulting phosphorylation of Mfn1 to mitochondrial morphology and bioenergetics and to the pathology associated with heart failure. Our study provides evidence that inhibition of excessive Mfn1-βIIPKC interaction and the resulting mitochondrial fragmentation and dysfunction are critical to heart failure-associated pathophysiology.

## Results

### βIIPKC activation mediates mitochondrial fragmentation in heart failure.
βIIPKC activation contributes to heart failure pathophysiology[14]. However, the molecular mechanisms involved in this process, including the identification of critical substrates that are phosphorylated by this pleiotropic enzyme are not known. As we demonstrated before[11], blocking protein–protein interaction between βIIPKC and its anchor protein (RACK1) with βII$_{V5-3}$ (the global inhibitor of βIIPKC[15]) improved isolated cardiomyocyte and whole heart contractility properties as well as cardiac remodeling in rats with myocardial infarction-induced heart failure (Fig. 1a–d; Supplementary Table 1). In these studies, rats were treated for six weeks with βII$_{V5-3}$ at 3 mg per Kg per day (a dose that we previously showed to be optimal[11]) in a sustained fashion using an Alzet pump implanted under the skin on the back of the animal. Treatment commenced after heart failure was established, 4 weeks after left descending coronary artery ligation to induce myocardial infarction (Fig. 1a). Control rats (sham group) were submitted to the same surgical procedure, but without left descending coronary artery ligation.

Using transmission electron microscopy, we noticed that the inhibition of global βIIPKC activity had a prominent impact on cardiac mitochondrial number and size in heart failure (Fig. 1e, f). Failing hearts accumulated smaller and spherical (fragmented) mitochondria compared to the sham group, resulting in increased mitochondrial number to size ratio (Fig. 1e, f). These changes correlated with increased outer mitochondrial membrane proteins, but not the electron transport chain subunits, in failing hearts (Supplementary Figure 1a-c). Sustained inhibition of global βIIPKC activity using the βII$_{V5-3}$ peptide at 3 mg per Kg per day re-established mitochondrial number to size ratio in rats with heart failure as compared to sham rats (Fig. 1e, f). These findings demonstrate that βIIPKC activation causes mitochondrial fragmentation in heart failure. Sustained global inhibition of βIIPKC activity had no impact on mitochondrial DNA levels and markers of mitophagy in heart failure (Supplementary Figure 1a, d, e). Neither cardiac function nor mitochondrial number to size ratio were affected by chronic treatment with peptide βII$_{V5-3}$ in Sham animals (Supplementary Figure 2).

Next, we demonstrated that only βIIPKC among different PKC isozymes accumulated in the mitochondrial fraction in failing rat hearts (Fig. 1g, h). RACK1, the βIIPKC anchoring protein[16] also accumulated in the mitochondrial fraction of failing hearts (Fig. 1g, h). Moreover, blocking βIIPKC-RACK1 interaction using βII$_{V5-3}$ peptide was sufficient to reduce βIIPKC levels in the mitochondrial fraction in heart failure (Fig. 1g, h). To determine whether βIIPKC is located at the outer mitochondrial membrane or inside mitochondria, we incubated isolated cardiac mitochondria with proteinase K (5 μg mL$^{-1}$) for 15 min in isotonic mitochondrial buffer. Proteinase K completely degraded βIIPKC and Tom20 (a protein located in the outer mitochondrial membrane), but not ALDH2 (an enzyme located in the matrix) (Fig. 1i). These findings demonstrated that, in failing rat hearts, βIIPKC accumulates and binds to RACK1, as a result of βIIPKC activation[17], in the outer mitochondrial membrane.

Next, we validated our in vivo findings using neonatal cardiac myocyte cultures. Treatment of neonatal cardiac myocytes with angiotensin II for 4 h caused mitochondrial fragmentation, which was prevented by a concomitant treatment with the global βIIPKC inhibitor, βII$_{V5-3}$ (Fig. 2a–c). The treatment also prevented mitochondrial accumulation of βIIPKC that was induced by either angiotensin II or ceramide (Fig. 2d); these stressors are hallmarks of heart failure in humans[18,19] that cause mitochondrial fragmentation in cardiomyocytes[20,21].

To examine the functional importance of βIIPKC on mitochondrial biology and cell survival, βIIPKC was silenced using selective small interfering RNA (siRNA, Fig. 2e). Silencing βIIPKC protected neonatal cardiac myocytes from H$_2$O$_2$-mediated oxidative stress-induced metabolic dysfunction and cytotoxicity (Fig. 2f). βII$_{V5-3}$ treatment and βIIPKC knockdown had a similar protective effect on neonatal cardiac myocytes, and the combination of genetic and pharmacological βIIPKC suppression had no additional protection on neonatal cardiac myocyte metabolism and toxicity as compared with either single intervention (Fig. 2e, f). βIIPKC knockdown (with siRNA) or

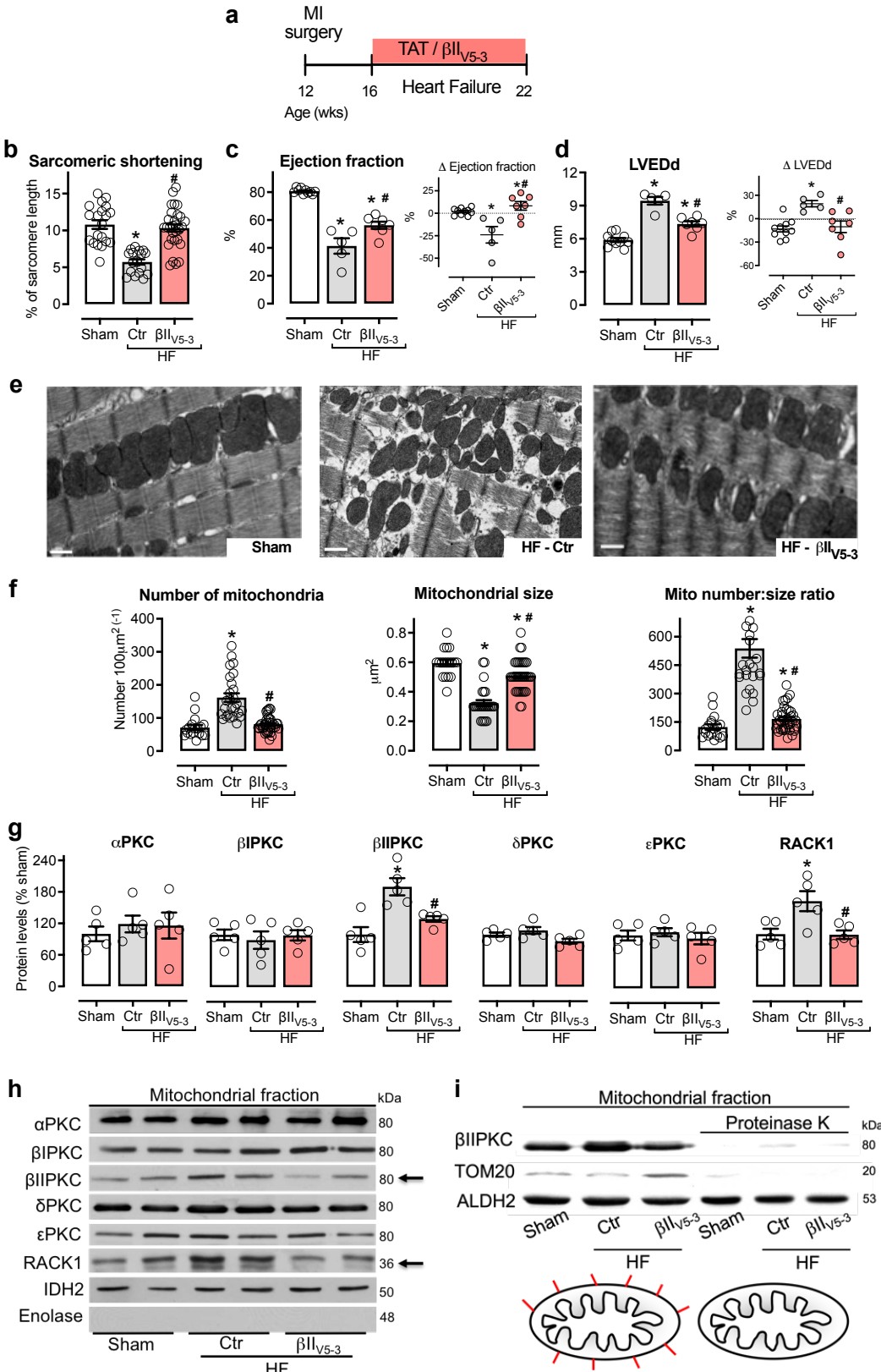

treatment with $\beta II_{V5-3}$ also protected adult cardiomyocytes from oxidative stress (a known trigger of mitochondrial fragmentation[22]) (Fig. 2g, h), similar to what we found in neonatal cardiomyocytes. These findings demonstrate that suppressing $\beta IIPKC$ is sufficient to protect both neonatal and adult cardiac

myocytes against oxidative stress-induced mitochondrial fragmentation and dysfunction.

We next determined whether $\beta IIPKC$ activation affects mitochondrial oxidative capacity in heart failure. To assess mitochondrial function, we measured oxygen consumption,

**Fig. 1** βIIPKC inhibition reduces mitochondrial fragmentation in heart failure. **a** Schematic panel of heart failure induction by myocardial infarction (MI) and the treatment protocol. Twelve-week-old male rats were subjected to MI-induced heart failure by left anterior descending coronary artery permanent ligation. Four weeks after MI induction, rats were treated with either the global βIIPKC-specific inhibitor (βII$_{V5-3}$) or with a control peptide (TAT, used to deliver βII$_{V5-3}$ into the heart). Peptide treatment was continuous (for 6 weeks) using an Alzet pump, delivering at a rate of 3 mg per Kg per day. **b** Sarcomeric shortening in isolated ventricular cardiomyocytes, **c** left ventricular ejection fraction and **d** LVEDd [left ventricular end-diastolic dimension] measured by echocardiography at the end of the experimental protocol, input: delta of measurements performed before and after treatment; **e** representative cardiac transmission electron micrographs (scale bar: 0.5 μm); **f** quantification of intermyofibrillar mitochondrial number and area in the transmission electron micrographs; **g** PKCs and RACK1 protein levels in cardiac mitochondrial fraction and **h** representative western blots; and **i** representative western blots of mitochondrial βIIPKC location in sham (white bars, $n = 8$), TAT-treated heart failure (HF-Ctr, gray bars, $n = 5$) and βII$_{V5-3}$-treated heart failure (HF-βII$_{V5-3}$, red bars, $n = 7$). Biochemical measurements were performed in the cardiac remote (viable) zone. Data are means ± SEM. *$P < 0.05$ vs. Sham rats. #$P < 0.05$ vs. HF-Ctr rats. One-way analysis of variance (ANOVA) with post-hoc testing by Duncan. For all the cardiac function studies, the observer was blinded to the experimental groups

absolute ($H_2O_2$) and relative ($H_2O_2$:$O_2$) hydrogen peroxide release in isolated mitochondria from heart failure rats. Failing hearts displayed a significant reduction in the efficiency of mitochondrial oxidative phosphorylation compared to hearts from sham operated rats, as measured by respiratory control ratio (RCR, state 3: state 4; Fig. 3a). This response was mainly due to a reduction of State 3 respiratory rate (Fig. 3b) and basal respiration (State 2, Supplementary Figure 3a), while respiration in the absence of oxidative phosphorylation (State 4) was unaffected in these failing rat hearts (Supplementary Figure 3a). Moreover, these failing hearts had elevated mitochondrial State 2 (basal, Supplementary Figure 3a) and State 3 (oxygen consumption rate maximized by the addition of ADP, Fig. 3c) $H_2O_2$ release compared to hearts from the sham group. These changes were maximized when $H_2O_2$ was normalized by oxygen consumption rates (Fig. 3d, Supplementary Figure 3a), providing evidence of a tight connection between mitochondrial oxygen consumption and reactive oxygen species release in heart failure.

We next determined the effect of the global βIIPKC inhibition on mitochondrial bioenergetics in heart failure. Sustained inhibition of βIIPKC activity with βII$_{V5-3}$ improved the efficiency of mitochondrial oxidative phosphorylation (Fig. 3a), mainly by preserving State 3 respiratory rate in heart failure (Fig. 3b). Moreover, mitochondria isolated from heart failure animals treated with the global βIIPKC inhibitor displayed reduced $H_2O_2$ and $H_2O_2$:$O_2$ release when compared to TAT-treated (control) heart failure animals (Fig. 3c, d). These findings suggest that inhibiting βIIPKC accumulation in the outer mitochondrial membrane is associated with the increase of mitochondrial oxygen consumption and reduced excessive mitochondrial $H_2O_2$ release in rats with heart failure. Next, we determined how βIIPKC affects mitochondrial biology in heart failure.

**βIIPKC interacts with and inhibits Mfn1 activity.** Mitochondrial number and size are tightly regulated by fusion and fission, in a process termed mitochondrial dynamics. Because heart failure is associated with excessive mitochondrial fragmentation, we hypothesized that elevated βIIPKC levels on the outer mitochondrial membrane resulted in phosphorylation of proteins that regulate fusion or fission. We first determined the levels and activity of the main GTPases involved in mitochondrial dynamics in the heart. Only the levels of Mfn1 (a mitochondrial fusion-related enzyme) were elevated in cardiac mitochondria during heart failure (Fig. 3e), yet its GTPase activity was significantly reduced as compared to the sham group (Fig. 3f), suggesting accumulation of dysfunctional Mfn1 in heart failure. βIIPKC inhibition by βII$_{V5-3}$ selectively re-established cardiac Mfn1 levels and activity in heart failure, without affecting the level or activity of mitofusin 2 (Mfn2, Fig. 3e, f).

Given that βIIPKC and Mfn1 accumulated on the outer mitochondria membrane during heart failure, and that blocking

global βIIPKC activity reduced mitochondrial Mfn1 levels and re-established the mitochondrial number to size ratio, we next determined whether βIIPKC and Mfn1 form a complex. Analysis of anti-Mfn1 immunoprecipitates by immunoblotting with anti-βIIPKC antibody demonstrated an increased association of βIIPKC with Mfn1 in cardiac mitochondria isolated from rats with heart failure (Fig. 3g, top panel). In reciprocal experiments, immunoblot analysis of anti-βIIPKC immunoprecipitates with anti-Mfn1 antibody confirmed the interaction (Fig. 3g, middle panel). No change in cardiac βIIPKC-Mfn2 interaction was seen in heart failure (Supplementary Figure 1f). Confirming a role for Mfn1 in βIIPKC association with the mitochondria, in Mfn1-null MEFs (mouse embryonic fibroblasts) mitochondrial βIIPKC levels were lower as compared to WT MEFs (Fig. 3h). However, Mfn1 is not essential for mitochondrial accumulation of βIIPKC upon stimulation with 10 nM phorbol ester 12-myristate 13-acetate (PMA, an activator of most PKC isozymes[11]).

Next, we determined the functional impact of βIIPKC on Mfn1. Immunoblot analysis of anti-Mfn1 immunoprecipitates with anti-serine/threonine antibody revealed an increased phosphorylation at the molecular size of Mfn1 in heart failure samples (Fig. 3g, middle panel) that was reduced by inhibiting βIIPKC activity with βII$_{V5-3}$. Co-immunoprecipitation experiments demonstrated that recombinant βIIPKC interacts with recombinant Mfn1 (Fig. 4a) and active βIIPKC (with PS/DG/$Ca^{+2}$), but not inactive βIIPKC or its alternative splicing βIPKC, phosphorylated Mfn1 [measured by incorporation of radiolabeled $P^{32}$-ATP] (Fig. 4b). Finally, as expected from our findings in whole hearts, active recombinant βIIPKC (and not active βIPKC) significantly reduced the GTPase activity of the recombinant Mfn1 in this in vitro assay (Fig. 4c).

Mass spectrometry analysis of the in vitro phosphorylation products identified S86, S284, and S290 in the Mfn1 GTPase domain to be the βIIPKC phosphorylated sites (Fig. 4d). A serine-to-alanine substitution demonstrated that S86A, but not S284A or S290A, was sufficient to protect MEFs against βIIPKC activation-induced cytotoxicity (Fig. 4e), suggesting S86 as a critical site to induce βIIPKC-mediated Mfn1 inactivation. Mfn1 S86 is conserved in many species and it is not present in Mfn2 (another critical mitofusin involved in mitochondrial dynamics; Fig. 4d). Collectively, these findings suggest that phosphorylation of Mfn1 by βIIPKC at S86 inhibits its GTPase activity and increases cytotoxicity.

**Design of selective inhibitor of Mfn1-βIIPKC association.** Our in vivo and in vitro data provide evidence that Mfn1 phosphorylation and consequent inhibition of its GTPase activity by βIIPKC is associated with impaired mitochondrial morphology and oxidative capacity. However, in addition to Mfn1, βIIPKC phosphorylates many other substrates. To determine whether inhibition of Mfn1-βIIPKC interaction alone is sufficient to

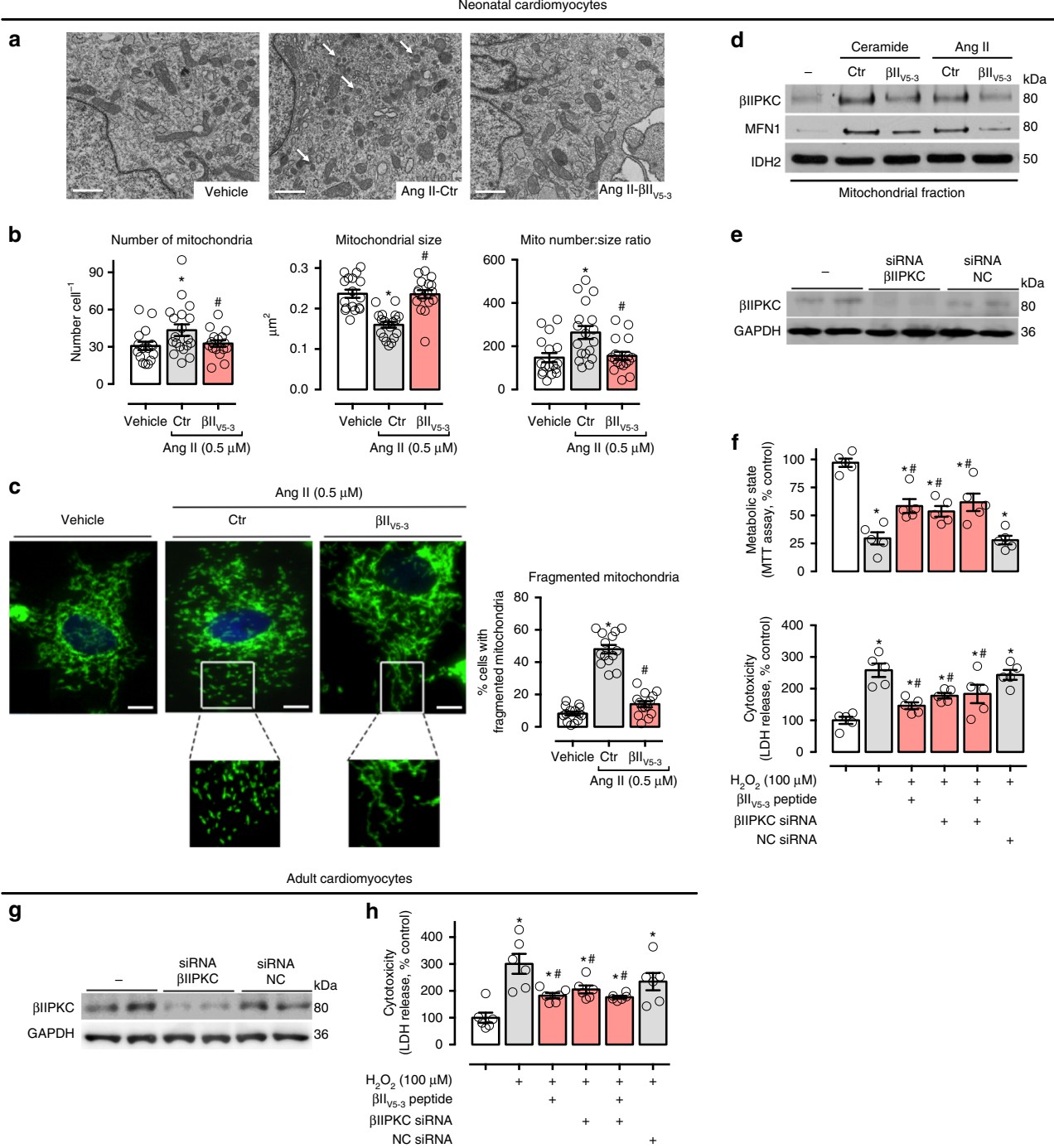

**Fig. 2** βIIPKC inhibition reduces mitochondrial fragmentation in cultured cardiomyocytes. **a** Representative transmission electron micrographs (white arrows indicate small mitochondria, scale bar: 1 μm) and **b** quantification of mitochondrial number and size in the transmission electron micrographs of neonatal rat cardiomyocytes in culture treated with TAT (carrier) or βII$_{V5-3}$ peptides for 30 min followed by incubation with angiotensin II (0.5 μM for 4 h, $n = 5$ per group). **c** Cells were then stained with anti-Tom20 antibody (green) and Hoechst stain and counted. Mitochondrial morphology was analyzed using a 63x oil immersion lens (scale bar: 5 μm). The analysis was done in a blinded fashion. The boxed area in each upper panel is enlarged under each micrograph. **d** Neonatal rat cardiomyocytes in culture were treated with TAT or βII$_{V5-3}$ peptides for 30 min followed by incubation with either C2-ceramide (40 μM) or angiotensin II (0.5 μM) for 4 h. After incubation, the mitochondrial levels of βIIPKC and Mfn1 were detected using specific antibodies (representative blot of three independent experiments). **e** Neonatal rat cardiomyocytes in culture were transfected with βIIPKC silence RNA (βIIPKC-siRNA) or control (NC-siRNA). **f** 48 h later cells were treated with TAT or βII$_{V5-3}$ peptides for 30 min followed by incubation with H$_2$O$_2$ (100 μM, 24 h). After incubation, cell viability and toxicity were measured by MTT assay and LDH release, respectively. **g** Adult rat cardiomyocytes in culture were transfected with βIIPKC-siRNA or NC-siRNA ($n = 6$ per group). **h** 24 h later cells were treated with TAT or βII$_{V5-3}$ peptides for 30 min followed by incubation with H$_2$O$_2$ (100 μM, 1 h). Cell toxicity was measured by LDH release. Data are means ± SEM. *$P < 0.05$ vs. control cells. #$P < 0.05$ vs. Ang II- or H$_2$O$_2$-treated cells. One-way analysis of variance (ANOVA) with post-hoc testing by Duncan

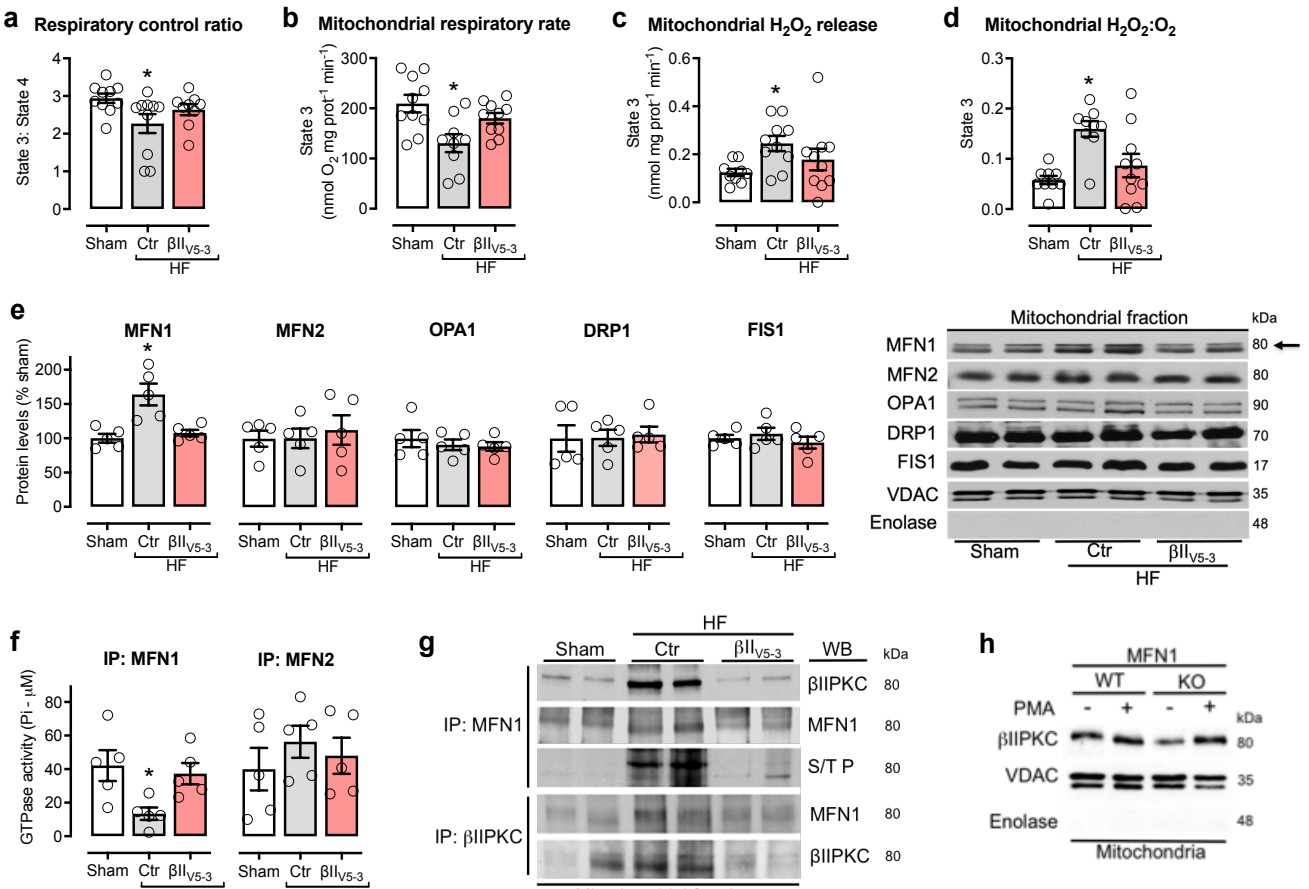

**Fig. 3** βIIPKC inhibition improves bioenergetics in failing hearts. **a** Mitochondrial respiratory control ratio, **b** State 3-dependent oxygen control rate, **c** absolute $H_2O_2$ release and **d** $H_2O_2$:$O_2$ in heart samples from sham (white bars, $n = 10$), TAT-treated heart failure (HF-Ctr, gray bars, $n = 10$) and βII$_{V5-3}$-treated heart failure (HF-βII$_{V5-3}$, red bars, $n = 10$). **e** Mfn1, Mfn2, Opa1, Drp1, and Fis1 protein levels and representative western blots; and **f** Mfn1 and Mfn2 GTPase activity in cardiac mitochondrial fraction ($n = 5$ per group) from sham, HF-Ctr and HF-βII$_{V5-3}$. **g** Cardiac mitochondrial Mfn1 and βIIPKC immunoprecipitate probed against anti-βIIPKC, Mfn1 and phosphorylated serine/threonine antibodies (representative blot of three independent experiments) from sham, HF-Ctr and HF-βII$_{V5-3}$ groups. Biochemical measurements were performed in the cardiac remote (viable) zone. These measurements were performed at the end of the experimental protocol. **h** Mfn1 wild type (WT, $n = 5$) and knockout (KO, $n = 5$) MEFs were treated with PMA [10 nM phorbol ester 12-myristate 13-acetate, for 30 min, an activator of most PKC isozymes[11]] and accumulation of mitochondrial βIIPKC was analyzed by western blot (representative blots of three independent experiments). Data are means ± SEM. *$P < 0.05$ vs. Sham rats. One-way analysis of variance (ANOVA) with post-hoc testing by Duncan

prevent accumulation of fragmented and dysfunctional mitochondria upon stress, we developed a peptide that selectively antagonizes Mfn1-βIIPKC association, termed SAMβA.

We have previously shown that two proteins that interact usually share short sequences of homology that represent sites of both inter- and intra-molecular interactions[17,23,24]. More recently, we described a similar rational approach to develop PKC inhibitors that selectively block interaction and phosphorylation of only one protein substrate of these multi-substrate kinases[6,7,25]. Using the same approach, we rationally designed SAMβA. First, we used sequence alignment software[26] to identify Mfn1-like sequence in βIIPKC (Fig. 5a).

The βIIPKC/Mfn1 homologous sequences, highlighted in color in Fig. 5a in the crystal structure of Mfn1 (5GOM) and βPKC (3PFQ), appear to be exposed α-helixes in the proteins and thus, likely accessible for protein–protein interaction (Fig. 5b). To help stabilize the α-helix structure, we added two amino acids at the C-terminus of each peptide. We reasoned that if these sequences are important for the protein–protein interaction between βIIPKC and Mfn1, they should be conserved across species. Indeed, RNAENFDRF is conserved in βIIPKC, but not in its alternative

splicing βIPKC (Fig. 5c, d). Conversely, NELENFTKQ is conserved in Mfn1, but not in the homologous Mfn2 protein (Fig. 5e, f). A BLAST search of the human genome identified four other proteins with homologous sequences. However, these sequences were not evolutionarily conserved (Fig. 5g; Supplementary Table 2), except in αPKC (Fig. 5g). However, mitochondrial αPKC levels were not changed in failing hearts compared to sham animals (Fig. 1g). Together these data suggest a specific role for RNAENFDRF for βIIPKC and Mfn1 protein–protein interaction.

Next, we synthesized the peptide RNAENFDRF (SAMβA) and conjugated it to the cell penetrating TAT protein-derived peptide, TAT$_{47-57}$[27], and determined its effect in protecting MEFs against oxidative stress-induced cytotoxicity. Four other peptides related to protein–protein interaction derived from either βIIPKC or Mfn1 were used as controls (P251–254, Supplementary Figure 4). Oxidative stress was induced by treating cells with $H_2O_2$. Only SAMβA (derived from βIIPKC), but not P255, derived from Mfn1, or P251–254 control peptides, was effective in mitigating oxidative stress-induced cytotoxicity (Fig. 5h, i upper panel). SAMβA also protected WT MEFs, but not Mfn1 knockout MEFs,

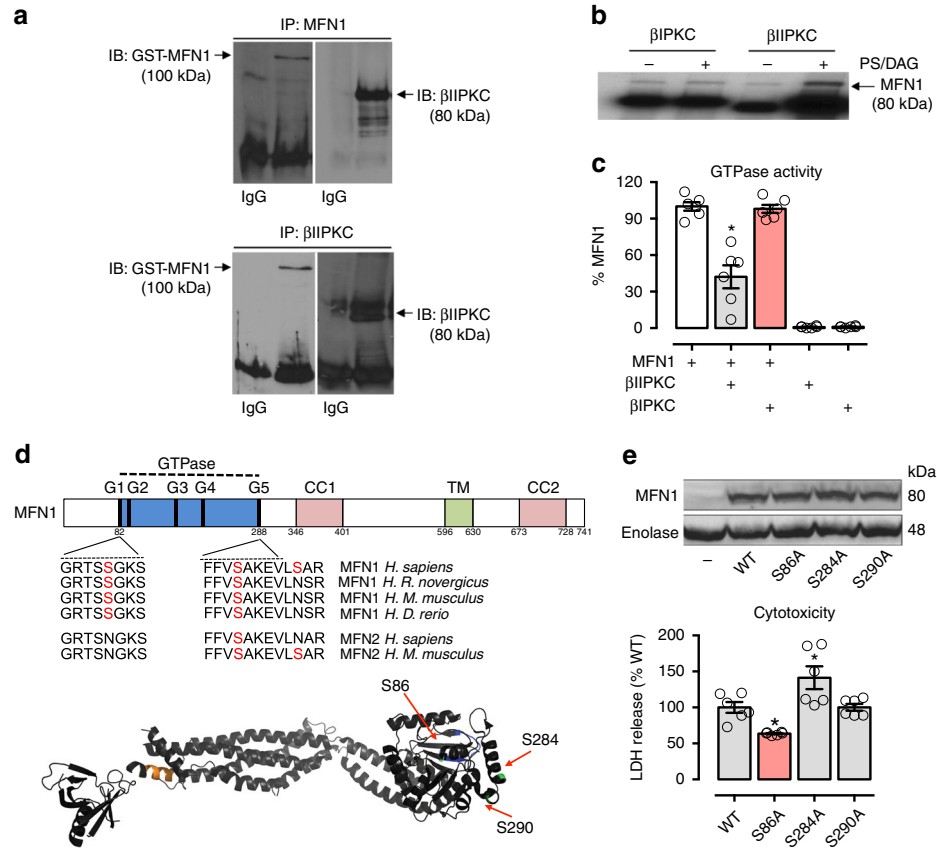

**Fig. 4** Characterization of Mfn1-βIIPKC protein–protein interaction. Recombinant Mfn1 (GST-Mfn1, 25 ng) was incubated with recombinant βIIPKC (GST-βIIPKC, 25 ng) in the presence and absence of phosphatidylserine and diacylglycerol (PS/DAG/$Ca^{2+}$, PKC activators) for 30 min at 37 °C. **a** Co-immunoprecipitates with anti-Mfn1 and βIIPKC were analyzed by western blot. **b** Mfn1 phosphorylation was evaluated using radioactive [$^{32}$P] $^{32}$P-ATP incorporation after incubation with either βIIPKC or its alternative splicing βIPKC in the presence and absence of classic PKC activators. Representative blots of three independent experiments. **c** GTPase activity of Mfn1 determined in the presence of active βIIPKC or its alternative splicing βIPKC ($n = 6$ per group). **d** Conservation of the βIIPKC phosphorylation in human orthologs of Mfn1 in rat, mouse, and zebrafish and Mfn2 in mouse. Ribbon presentation of the 3D structure of Mfn1 phosphorylation at the βIIPKC site (PDB: 5GOM). **e** Individual replacement of serine by alanine at residues 86, 284, and 290 in Mfn1 knockout MEFs. Cells were transfected with the following constructs: WT, S86A, S284A and S290A. 48 h after transfection cells were incubated with $H_2O_2$ (100 μM, 24 h) and measured cytotoxicity (LDH release, $n = 6$ per group). Data are means ± SEM. *$P < 0.05$ vs. control. One-way analysis of variance (ANOVA) with post-hoc testing by Duncan

from antimycin A, an inhibitor of the mitochondrial electron transport chain (Fig. 5h, i lower panel).

Given that mitochondrial Mfn1 phosphorylation and inhibition by βIIPKC is associated with accumulation of fragmented/dysfunctional mitochondria and oxidative stress in failing hearts, we next determined the effect of the SAMβA peptide on mitochondrial morphology in cardiac myocytes. Treatment of cultured neonatal cardiac myocytes with SAMβA prevented mitochondrial fragmentation triggered by angiotensin II (Fig. 5j). SAMβA reduced the excessive association of βIIPKC with Mfn1 induced by angiotensin II in cardiac myocytes, as measured by co-immunoprecipitation (Fig. 5k). Finally, the treatment of cardiac neonatal myocytes with SAMβA had a better effect in protecting against $H_2O_2$-induced metabolic dysfunction when compared to the global βIIPKC inhibitor, βII$_{V5-3}$ (Fig. 5l).

**SAMβA treatment improves heart failure outcome in rats**. To investigate the contribution of Mfn1-βIIPKC interaction to the pathophysiology of heart failure, we delivered the SAMβA peptide (or its control peptide, TAT) to rats with myocardial infarction-induced heart failure; the peptides were delivered in a sustained fashion at 3 mg per kg per day, using subcutaneously implanted

Alzet pumps. Treatments were commenced 4 weeks after myocardial infarction and lasted 6 weeks (from 4 to 10 after inducing myocardial infarction; Fig. 6a). This 6-week treatment with SAMβA resulted in a significant increase of cardiac function (measured by ventricular ejection fraction) compared to TAT-treated (control) heart failure animals (Fig. 6b). SAMβA treatment also decreased left ventricular end-diastolic diameter (LVEDd) as compared to control heart failure group (Fig. 6c; Supplementary Table 3). The better cardiac function following SAMβA treatment was likely due to improved integrity of cardiac myofibril and mitochondria structures, as evidenced by transmission electron microscopy (Fig. 6d); sustained SAMβA treatment prevented the accumulation of smaller and spherical (fragmented) mitochondria seen in the control rats with heart failure. Sustained SAMβA delivery had no impact on blood pressure in rats with heart failure (Supplementary Table 3).

Sustained inhibition of Mfn1-βIIPKC interaction using SAMβA re-established mitochondrial number to size ratio in heart failure compared to sham group (Fig. 6d, e). Six weeks of treatment with SAMβA also improved mitochondrial oxidative phosphorylation efficacy in heart failure (Fig. 6f), mainly by preserving State 2 (Supplementary Figure 3b) and State 3 respiratory rates (Fig. 6g). In addition, rats with established heart

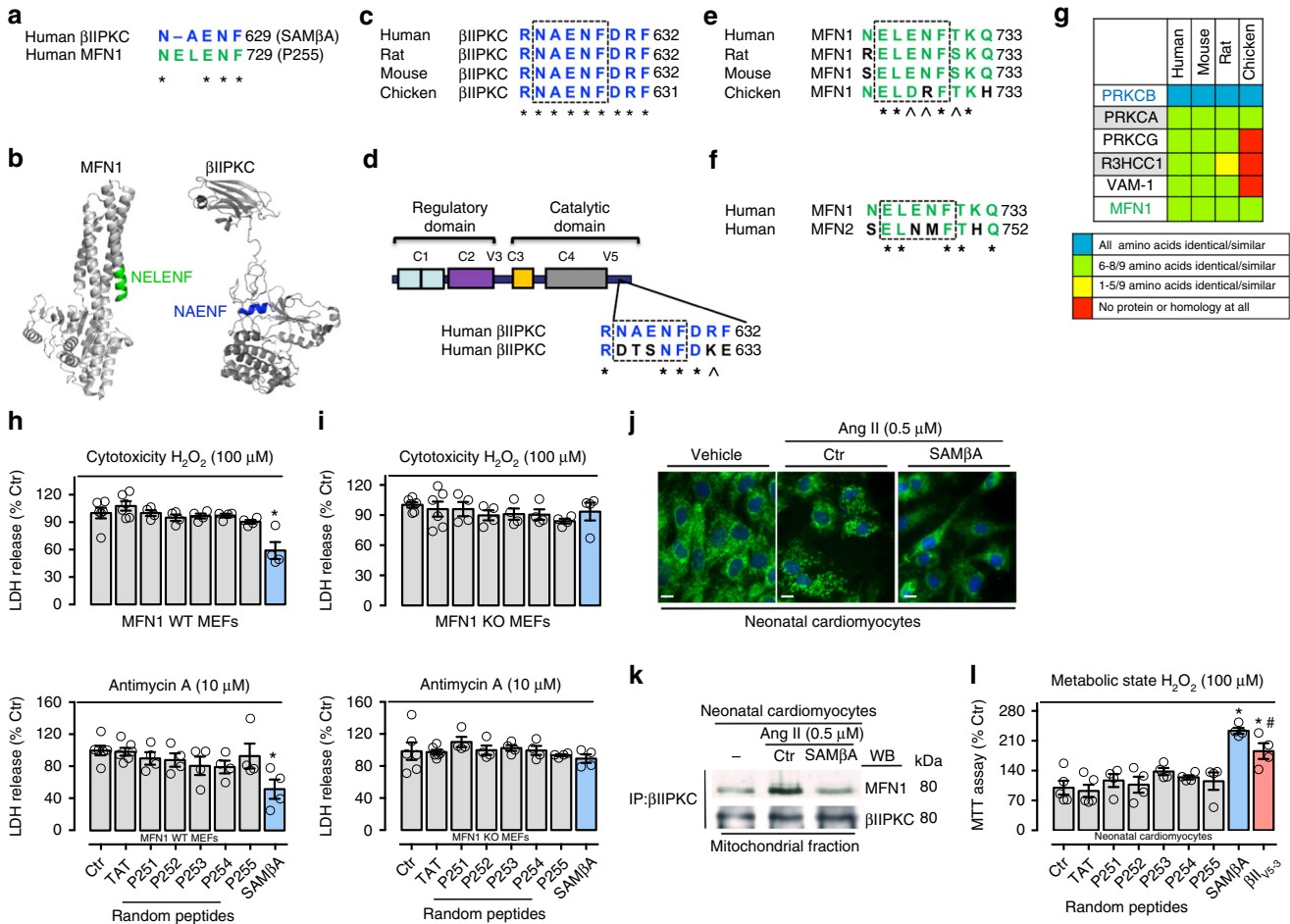

**Fig. 5** Rational design of peptide that selectively inhibits Mfn1 and βIIPKC interaction. **a** Sequence alignment of human βIIPKC and Mfn1 identified a short sequence of homology, RNAENF/NELENF. **b** NELENF in Mfn1 (PDB: 5GOM) and RNAENF in the V5 domain of βIIPKC (PDB: 3PFQ) are exposed and available for protein−protein interactions (see colored structures). Note that the alternatively spliced βIPKC varies from βIIPKC only in the V5 domain[15]. **c** Conservation of RNAENFDRF sequence in βIIPKC in a variety of species. **d** RNAENFDRF sequence is not present in the V5 domain of βIPKC, a βIIPKC alternative spliced form. **e** Conservation of NELENFTKQ in Mfn1 in a variety of species. **f** NELENFTKQ sequence is not present in Mfn2. **g** RNAENFDRF sequence is found in 6 human proteins. Heat-map of the RNAENFDRF conservation in orthologous of these proteins shows RNAENFDRF conservation only in βIIPKC. (Further information about all the proteins is given in the Supplementary Table 2). *Denotes identity, and ˄ denotes homology. **h** Mfn1 wild type (WT, $n = 4–6$) and **i** Mfn1 knockout (KO, $n = 4–6$) MEFs were treated with TAT, p251–255 or SAMβA peptides for 30 min followed by incubation with either $H_2O_2$ (100 μM, 24 h) or Antimycin A (10 μM, 24 h). After incubation, cytotoxicity was measured by LDH release. **j** Neonatal rat cardiomyocytes in culture were treated with TAT or SAMβA for 30 min followed by incubation with angiotensin II (0.5 μM for 4 h, $n = 5$ per group). Cells were then stained with anti-Tom20 antibody (green) and Hoechst stain and counted. Mitochondrial morphology was analyzed using a ×63 oil immersion lens (scale bar: 10 μm). **k** Co-immunoprecipitates with anti-βIIPKC were analyzed in the mitochondrial fraction by western blot (representative blots of three independent experiments). **l** Neonatal rat cardiomyocytes in culture were treated with TAT, P251–255, SAMβA or βII$_{V5-3}$ peptides for 30 min followed by incubation with $H_2O_2$ (100 μM for 24 h, $n = 4–5$ per group). After incubation, cell viability was measured by MTT assay. Data are means ± SEM. *$P < 0.05$ vs. Ctr. #$P < 0.05$ vs. SAMβA-treated cells. One-way analysis of variance (ANOVA) with post-hoc testing by Duncan

failure that were treated with the SAMβA for 6 weeks presented reduced cardiac mitochondrial $H_2O_2$ and $H_2O_2:O_2$ release when compared to control heart failure rats (Fig. 6h, i; Supplementary Figure 3b). Of interest, sustained SAMβA treatment was sufficient to reduce apoptotic cell death in failing hearts as compared to that in hearts from vehicle-treated rats (Supplementary Figure 3c). Therefore, sustained blockage of Mfn1-βIIPKC interaction was critical to protect against accumulation of fragmented/dysfunctional mitochondria with an impact on cell death in heart failure.

Next, we tested the effectiveness of SAMβA on reducing Mfn1-βIIPKC interaction in failing hearts. Immunoblot analysis of anti-βIIPKC immunoprecipitates with anti-Mfn1 and anti-serine/threonine antibodies demonstrated increased Mfn1-βIIPKC association and phosphorylation at the molecular size of Mfn1 in heart failure samples that were reduced by SAMβA (Fig. 6j, left panel).

The selectivity of SAMβA as a Mfn1-βIIPKC association and phosphorylation inhibitor was also tested by determining its impact on βIIPKC-troponin I association. Troponin I is a well-known βIIPKC substrate in heart failure[28]. Immunoblot analysis of anti-βIIPKC immunoprecipitates with anti-troponin I and anti-phosphorylated troponin I antibodies demonstrated an elevated βIIPKC-troponin I interaction and phosphorylation at the molecular size of troponin I in heart failure samples that were not affected by SAMβA (Fig. 6j, right panel). These findings provide evidence that SAMβA selectively targets the elevated Mfn1-βIIPKC interaction in heart failure and not another βIIPKC substrate.

Finally, we compared the benefit of treatments with βII$_{V5-3}$ (a global βIIPKC inhibitor) relative to SAMβA (a selective Mfn1-βIIPKC protein–protein interaction inhibitor) in improving ventricular function and remodeling in heart failure. Both

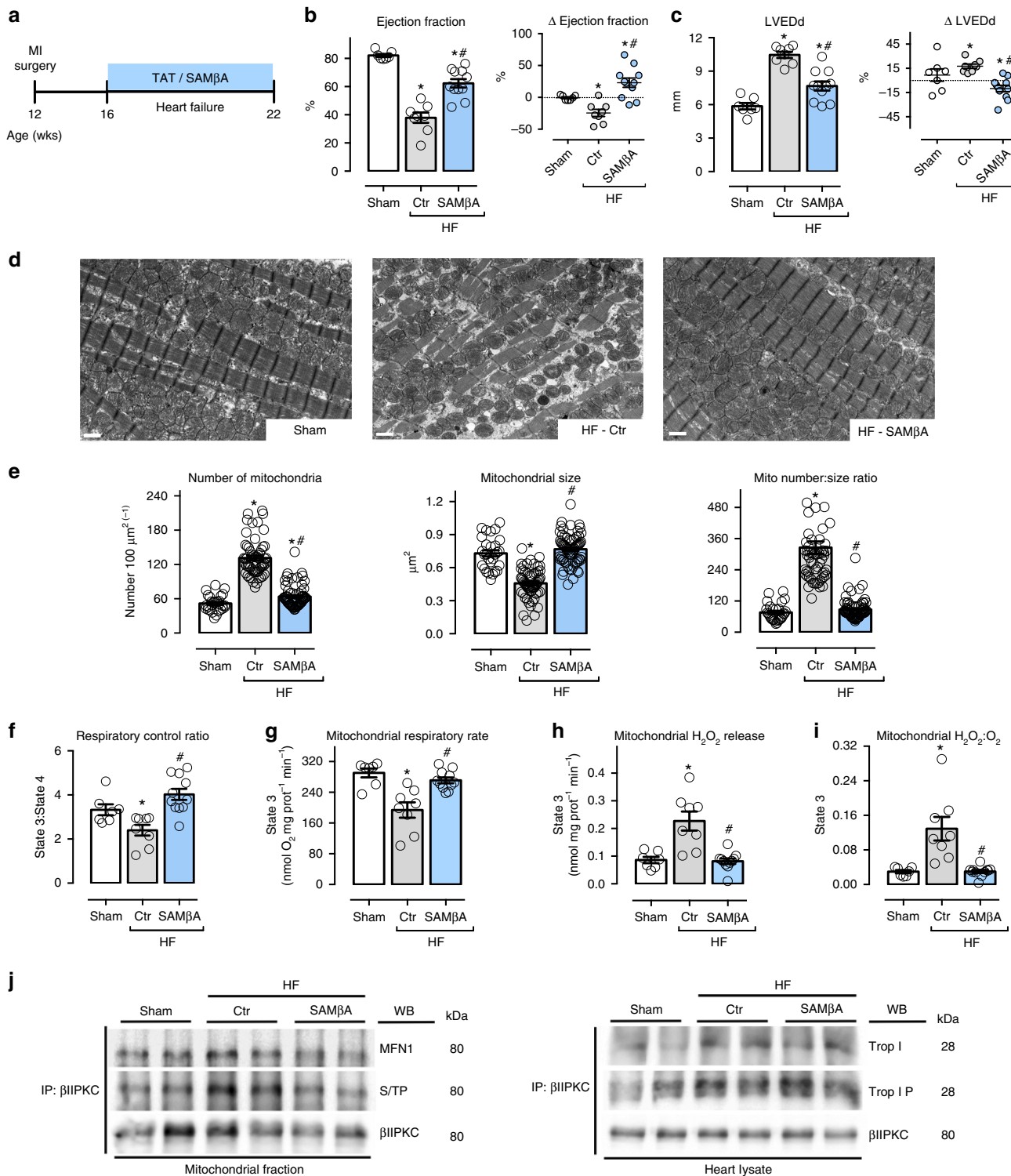

treatments prevented the progression of heart failure, measured by the drop in left ventricular ejection fraction and increase in cardiac dilation over time (before vs. after treatment, Fig. 7a, b). However, only the treatment with SAMβA resulted in a significant improvement in left ventricular ejection fraction when compared to week 4 (before treatment, Fig. 7a, b).

## Discussion

There is much interest in the role of mitochondrial dynamics in cardiac physiology and pathology. However, the upstream

molecular mechanisms affecting mitochondrial morphology in cardiac diseases are largely unknown. The preclinical studies described here suggest that βIIPKC activation contributes to heart failure by inhibiting Mfn1, therefore causing accumulation of fragmented/dysfunctional mitochondria, most likely resulting from impaired mitochondrial fusion.

Our previous studies demonstrated that βIIPKC, a serine-threonine protein kinase C isozyme, is elevated in human failing hearts[11] and plays a critical role in the progression of heart failure in rodents[12]. βIIPKC activation was also previously demonstrated to be involved in changes in mitochondrial morphology in MEFs upon

**Fig. 6** SAMβA treatment reduces mitochondrial fragmentation in heart failure. **a** Schematic panel of heart failure induction by myocardial infarction (MI) and the treatment protocol. Twelve-week-old male rats were subjected to MI-induced heart failure by left anterior descending coronary artery permanent ligation. Four weeks after MI induction, the rats were treated with the either SAMβA peptide (a selective antagonist of Mfn1-βIIPKC association) or with a control peptide (TAT, used to deliver SAMβA into the heart). Peptide treatment was continuous for 6 weeks using an Alzet pump delivery at 3 mg per Kg per day. **b** Left ventricular ejection fraction and **c** LVEDd [left ventricular end-diastolic dimension] measured by echocardiography at the end of the experimental protocol, input: delta of measurements performed before and after treatment; **d** representative cardiac transmission electron micrographs (scale bar: 1 μm); **e** quantification of mitochondrial number and area in the transmission electron micrographs, the analysis was done in a blinded fashion; **f** mitochondrial respiratory control ratio, **g** state 3-dependent oxygen control rate, **h** absolute $H_2O_2$ release and **i** $H_2O_2:O_2$ in heart samples from sham (white bars, $n = 7$), TAT-treated heart failure (HF-Ctr, gray bars, $n = 8$), and SAMβA-treated heart failure (HF-SAMβA, blue bars, $n = 11$). **j** Cardiac mitochondrial βIIPKC immunoprecipitate probed against anti-βIIPKC, Mfn1 and phosphorylated serine/threonine antibodies; and βIIPKC immunoprecipitate from heart lysate probed with anti-βIIPKC, troponin I and phosphorylated troponin I antibodies (representative blot of three independent experiments) from sham, HF-Ctr and HF-SAMβA groups. Biochemical measurements were performed in the cardiac remote (viable) zone. These measurements were performed at the end of the experimental protocol. Data are means ± SEM. *$P < 0.05$ vs. Sham rats. #$P < 0.05$ vs. HF-Ctr rats. One-way or two-way analyses of variance (ANOVA) with post-hoc testing by Duncan. For all the cardiac function studies, the observer was blinded to the experimental groups

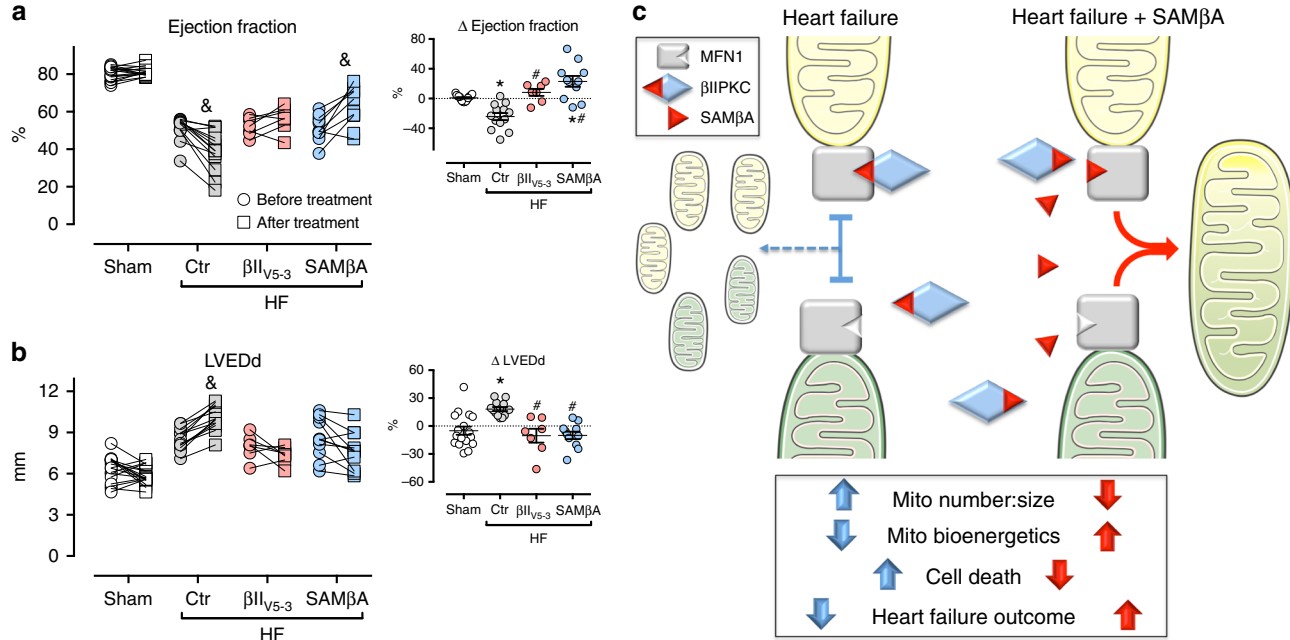

**Fig. 7** Sustained SAMβA treatment results in better heart failure outcome. **a** Left ventricular ejection fraction and **b** LVEDd (left ventricular end-diastolic dimension) measured by echocardiography before and after treatment; input: delta of measurements performed before and after treatment, in animals from sham (white bars, $n = 7$), TAT-treated heart failure (HF-Ctr, gray bars, $n = 9$), βII$_{V5-3}$-treated heart failure (HF-βII$_{V5-3}$, red bars, $n = 7$), and SAMβA-treated heart failure groups (HF-SAMβA, blue bars, $n = 11$). Data are means ± SEM. *$P < 0.05$ vs. Sham rats. #$P < 0.05$ vs. HF-Ctr rats. &$P < 0.05$ vs. before treatment. One-way or two-way analyses of variance (ANOVA) with post-hoc testing by Duncan. For all the cardiac function studies, the observer was blinded to the experimental groups. **c** Proposed model for the SAMβA-mediated cardioprotection in post-myocardial infarction-induced heart failure

oxidative conditions[29]. Here we identified at least one downstream βIIPKC substrate that mediates cardiac pathophysiology. We demonstrated that during heart failure, βIIPKC translocated to the mitochondrial outer membrane, where it is bound to and phosphorylated Mfn1. The phosphorylation of Mfn1 by βIIPKC at S86 located in the GTPase domain in Mfn1, correlated with a reduction of Mfn1 GTPase activity, but also correlated with an increase in Mfn1 levels at the mitochondria. This is in contrast to Mfn1 phosphorylation by ERK, which also induces mitochondrial fragmentation in MEFs[30], but it drives Mfn1-increased clearance thorough ubiquitin proteasome system[31,32]. The accumulation of βIIPKC-phosphorylated Mfn1 could be due to the site of phosphorylation and or because failing hearts have impaired both proteasomal proteolytic activity[11] and autophagy[33]. Regardless of the mechanism, βIIPKC phosphorylation of Mfn1 impairs its GTPase activity, which is critical for mitochondrial fusion activity[34]. Disruption of Mfn1 activity leads to metabolic dysfunction[35].

Finally, we confirmed that Mfn1 phosphorylation and inhibition by βIIPKC alone is a major contributor to the pathophysiology associated with heart failure in this post-myocardial infarction heart failure rat model, using SAMβA, a rationally designed selective antagonist of Mfn1-βIIPKC association. This inhibitory peptide is composed of a 9-amino-acid peptide (RNAENFDRF; βIIPKC$_{624-632}$) conjugated to the cell permeable peptide TAT$_{47-57}$[36]. A 6-week treatment of myocardial infarction-induced heart failure rats with SAMβA, starting 4 weeks after myocardial infarction at a time when ventricular remodeling and cardiac dysfunction were present, re-established mitochondrial morphology, bioenergetics, and redox balance. In addition, SAMβA had a superior effect on cardiac function, left ventricular compliance, and diastolic function compared to the global βIIPKC inhibitor, βII$_{V5-3}$.

Our previous studies showed that preventing accumulation of fragmented mitochondria using other pharmacological means is

also beneficial. A peptide inhibitor of phosphorylation of Drp1 by δPKC, δV1-1 or an inhibitor of Drp1 interaction with Fis1, P110, protect neurons and cardiomyocytes from acute ischemic stress[37–39] and a rationally designed peptide activator of Mfn2 corrects mitochondrial fusion defects in a model of Charcot-Marie-Tooth model in neurons[40]. Therefore, the molecular basis of increased mitochondrial fragmentation can be diverse, depending on the disease. However, in all these cases, correcting the mitochondrial morphology improves outcome, by correcting mitochondrial functions and ATP production. We suggest that the improvement in mitochondrial functions is critical for cellular health and that the benefit we saw with sustained treatment using SAMβA on reversal of the cardiac pathological remodeling and dysfunction induced by post-myocardial infarction is due to increased repair ability of the cells. From a clinical standpoint, future experiments checking the effectiveness of SAMβA in preventing the development of heart failure by starting the treatment immediately after myocardial infarction are needed. It is also important to validate the cardiac benefits of SAMβA by looking at different models of cardiomyopathy and heart failure.

In conclusion, our findings of increased mitochondrial fragmentation in failing hearts and its dependence on βIIPKC activation suggest that βIIPKC is a regulator of mitochondrial dynamics through Mfn1 phosphorylation, at least under pathological conditions. Our data also suggest that correcting impaired mitochondrial dynamics, in general, and selectively inhibiting Mfn1-βIIPKC interaction with drugs, such as SAMβA, may provide useful treatments for patients with established heart failure.

## Methods

**Sequence alignments**. Sequences from different species were aligned using FASTA server[41], using the following βIIPKC proteins: human (P05771-2), mouse (55977813), rat (gi66724), and chicken (A0A1D5PUY5). αPKC proteins: human (P17252), mouse (P20444), rat (P05696), and chicken (Q5F3X1). γPKC proteins: human (P05129), mouse (P63318), and rat (P63319). R3HCC1 proteins: human (Q9Y3T6), mouse (Q8BSI6), rat (Q9EQN5), and chicken (118101322). VAM-1 proteins: human (Q9NZW5), mouse (Q9JLB0), and rat (201066397). MFN1 proteins: human (26251799), mouse (34784646), rat (20376820), and chicken (F1NV08). MFN2 proteins: human (189083768) and βIPKC proteins: human (P05771-1).

**Animals**. The animal protocols were approved by the Stanford University Institutional Animal Care and Use Committee (Protocol ID: 14746) and by the Ethical Committee of Biomedical Sciences Institute of University of São Paulo (2013/134). This study was conducted in accordance with the Guide for the Care and Use of Laboratory Animals published by the US-National Institutes of Health and with the ethical principles in animal research adopted by the Brazilian Society of Laboratory Animal (www.cobea.org.br). The study with βIIV5-3 treatment in male rats included three groups: sham, $n = 10$; TAT-treated (control) heart failure, $n = 10$; βIIV5-3-treated heart failure, $n = 10$. The study with SAMβA treatment included sham, $n = 7$; TAT-treated (control) heart failure, $n = 8$ and SAMβA-treated heart failure, $n = 11$. No animals were excluded from the study. Sham and myocardial infarction surgery were done on the same day, usually 10 rats per day. Four weeks after myocardial infarction, rats were randomized to drug treatments.

**Myocardial infarction-induced heart failure**. We have chosen this rat model since myocardial infarction is the underlying etiology of heart failure in nearly 70% of patients[42]. Myocardial infarction was induced by a permanent ligation of the left anterior descending coronary artery in Wistar normotensive male rats at 12 weeks of age[43]. Rats were anesthetized intraperitoneally with ketamine (50 mg Kg⁻¹) and xylazine (10 mg Kg⁻¹), endotracheally intubated, and mechanically ventilated with room air (respiratory rate of 60–70 breaths per minute and tidal volume of 2.5 mL). Left thoracotomy between the fourth and fifth ribs was performed and the left anterior descending coronary artery was permanently ligated[44]. Four weeks later, echocardiography (Acuson Sequoia, 14 MHz) to evaluate fractional shortening was performed and rats showing heart failure symptoms were included in the study. Heart failure was defined when animal presented pathological cardiac remodeling accompanied by left ventricle dysfunction and cardiac dilation, according to the Guidelines of American Heart Association[45]. Left thoracotomy with equal procedure duration, but without left anterior descending coronary artery ligation, was undertaken in the sham group.

**Study design**. Heart failure animals were treated between the ages of 16 and 22 weeks. Rats were randomized to the treatment groups. Each group was treated with TAT$_{47–57}$-βII$_{V5-3}$ peptide (global βIIPKC inhibitor, 3 mg per Kg per day), TAT$_{47–57}$-SAMβA peptide Mfn1-βIIPKC inhibitor, 3 mg per Kg per day) or equimolar concentration of TAT$_{47–57}$-carrier peptide, using Alzet osmotic pumps under the skin on the back of the animal, and were replaced every 2 weeks. The sham group was subjected to TAT$_{47–57}$-carrier peptide treatment, as a negative control. At the end of the protocol, cardiac function was re-evaluated by observer blinded to the treatment groups. Forty-eight hours later, all rats were euthanized by decapitation for other analyses. Hearts were dissected into infarct and viable zones using the following criteria. The infarct zone was the left ventricular wall that was thinning and included visible infarct scar, and the viable (remote) zone was the part of the left ventricular wall that was reddish-brown and with normal thickness. All biochemical measurements were performed in the left ventricular viable (remote) zone.

**Cardiovascular measurements**. Systolic blood pressure was determined non-invasively, using a computerized tail-cuff system (BP-2000, Visitech System). Non-invasive cardiac function evaluation was performed by M-mode echocardiography in anesthetized (isoflurane 3%) rats. Briefly, rats were positioned in the supine position with front paws wide open and ultrasound transmission gel was applied to the precordium. Transthoracic echocardiography was performed using an Acuson Sequoia model 512 echocardiographer (SIEMENS) equipped with a 14-MHz linear transducer. Left ventricle systolic function was estimated by ejection fraction (EF) and fractional shortening (FS) as follows: EF (%) = [(LVEDD³−LVESD³) LVEDD³⁻¹]×100 and FS (%) = [(LVEDD—LVESD) LVEDD⁽⁻¹⁾]×100, where LVEDD is the left ventricular end-diastolic diameter, and LVESD is the left ventricular end-systolic diameter. The observer was blinded to the treatment groups.

**Isolation of adult rat cardiomyocytes**. Ten weeks after MI surgery, control, vehicle-treated HF and βII$_{V5-3}$-treated HF rats were injected with heparin (2000 U kg⁻¹ IP), anesthetized with sodium pentobarbital (100 mg kg⁻¹ IP) and euthanized by decapitation. The hearts were rapidly excised and then perfused with low-Ca²⁺ solution 1 (100 mM NaCl, 10 mM KCl, 1.2 mM KH₂PO₄, 5 mM MgSO₄, 20 mM glucose, 50 mM taurine, 10 mM HEPES, and 100 μM CaCl₂), then with digestion solution containing low-Ca²⁺ solution 1, collagenase (0.5 mg mL⁻¹, Worthington Type 1 A), and protease type XIV (0.04 mg mL⁻¹; Sigma). Following perfusion, the ventricles were cut into fragments (2–5 mm³) in digestion solution. The cell suspension was then filtered through a nylon sieve and centrifuged for 1 min (at 300–400 × g) at room temperature. Cell pellets were resuspended in solution 1 containing 125 mg BSA and 500 μM CaCl₂.

**Cardiomyocyte shortening and relengthening**. Cell contraction properties of adult cardiomyocytes were evaluated with a video-based sarcomere spacing acquisition system (SarcLen, IonOptix)[46]. Changes in sarcomere length were recorded and analyzed using IonWizard software (IonOptix). Sarcomeric shortening was determined under basal conditions.

**Mitochondrial isolation**. Cardiac samples from remote area were minced and homogenized in isotonic mitochondrial buffer (300 mM sucrose, 10 mM Hepes, 2 mM EGTA, pH 7.2, 4 °C) containing 0.1 mg mL⁻¹ of type I protease (bovine pancreas, Sigma P4630) to release mitochondria from within muscle fibers and later washed in the same buffer in the presence of 1 mg mL⁻¹ bovine serum albumin[47]. The suspension was homogenized in a 40 mL tissue grinder and centrifuged at 950 × g for 5 min. The resulting supernatant was centrifuged at 9500 × g for 10 min. The mitochondrial pellet was washed, resuspended in isolation buffer and submitted to a new centrifugation (9500 × g for 10 min). The mitochondrial pellet was washed and the final pellet was resuspended in a minimal volume of isolation buffer.

**Mitochondrial function**. All experiments with isolated mitochondria (0.125 mg mitochondrial protein mL⁻¹) were monitored in experimental buffer containing 125 mM sucrose, 65 mM KCl, 10 mM Hepes, 2 mM inorganic phosphate, 2 mM MgCl₂, 100 μM EGTA, and 0.01% BSA, pH 7.2 and were performed in the presence of succinate, malate, and glutamate substrates (2 mM of each) with continuous stirring at 37 °C[48]. Mitochondrial O₂ consumption was monitored using a computer-interfaced Clark-type electrode (OROBOROS Oxygraph—2k). ADP (1 mM—Amresco 0160) was added to induce State 3 respiratory rates. A subsequent addition of oligomycin (1 μg mL⁻¹—Sigma 4876) was used to determine State 4 rates. Respiratory control ratios were calculated by the State 3:State 4 ratio. Additionally, 0.1 mM FCCP [Carbonyl cyanide 4-(trifluoromethoxy) phenylhydrazone—Enzo BML-CM120] was added in order to evaluate O₂ consumption in uncoupled mitochondria[48]. Mitochondrial H₂O₂ release was measured using Amplex Red (25 μM—Molecular Probes A12222)-horseradish peroxidase (0.5 U mL⁻¹– Sigma P8125) system[48]. Amplex Red is oxidized in the presence of extramitochondrial horseradish peroxidase bound to H₂O₂, generating resorufin, which can be detected using a fluorescence spectrophotometer ($\lambda_{ex} = 563/\lambda_{em} = 587$ nm) (F-2500 Hitachi—Hitachi). To estimate H₂O₂ release during State 3 and State 4 respiratory rates, and in uncoupled mitochondria, we added ADP (1 mM),

oligomycin (1 µg mL$^{-1}$), and FCCP (0.1 mM), respectively. Calibration was conducted by adding $H_2O_2$ at known concentrations ($A_{240} = 43.6$ M$^{-1}$.cm$^{-1}$) to the experimental buffer.

**Immunoblotting.** Cardiac samples were subjected to SDS-PAGE in polyacrylamide gels (6–15%) depending upon protein molecular weight. After electrophoresis, proteins were electrotransferred to nitrocellulose membranes. Equal gel loading and transfer efficiency were monitored using 0.5% Ponceau S staining of blot membrane. Blotted membrane was then blocked (5% BSA, 10 mM Tris-HCl; pH 7.6), 150 mM NaCl, and 0.1% Tween 20) for 2 h at room temperature and then incubated overnight at 4 °C with specific antibodies against Drp1 (611113, dilution 1:1000) from BD Biosciences; GAPDH (RGM2, dilution 1:10000) from Advanced Immunochemical; Mfn1 (H00055669-M04, dilution 1:1000), Mfn2 (H00009927-M01, dilution 1:1000), Opa1 (H00004976-M01, dilution 1:1000), and IDH2 (H00003418-M01, dilution 1:1000) from Abnova; ATP5A (Ab14748, dilution 1:1000), NDUFA9 (Ab14713, dilution 1:1000), Ubiquinol (Ab110252, dilution 1:1000), and Troponin I (Ab47003, dilution 1:1000) from Abcam; SQSTM1/p62 (5114, dilution 1:1000) and phospho-Troponin I (serine 23/24–4004, dilution 1:1000) from Cell Signaling Technology; Optineurin (10837-1-AP, dilution 1:1000) and NDP52-CALCOCO2 (12229-1-AP, dilution 1:1000) from Proteintech; Fis1 (210-907-R100, dilution 1:1000) from Enzo Life Sciences and αPKC (sc-208, dilution 1:1000), βIPKC (sc-209, dilution 1:1000), βIIPKC (sc-210, dilution 1:500), δPKC (sc-213, dilution 1:500), εPKC (sc-214, dilution 1:500), RACK1 (sc-10775, dilution 1:500), Tom20 (sc-11021, dilution 1:500), VDAC (sc-32063, dilution 1:500), Enolase (sc-15343, dilution 1:500), ALDH2 (sc-100496, dilution 1:500), and Mfn1 (sc-50330, dilution 1:500) from Santa Cruz Biotechnology. Positive and negative controls for the detection of Mfn1 are shown in Supplementary Figure 4b. Binding of the primary antibody was detected with the use of peroxidase-conjugated secondary antibodies (rabbit, mouse or goat for 2 h at room temperature, dilution 1:10,000) and developed using enhanced chemiluminescence detected by autoradiography. Quantification analysis of blots was performed with the use of Image J software (Image J Corporation based on NIH image). The protein contents were quantified by the Bradford method[49]. Samples were normalized to relative changes in housekeeping proteins [GAPDH (heart lysate), VDAC (mitochondrial fraction) or IDH2 (mitochondrial fraction)] and expressed as the percent of control. Original blot data are available in the Supplementary information (Supplementary Figures 5–8).

**Immunoprecipitation.** Cardiac mitochondrial fraction (0.25 mg protein) and lysate (0.5 mg protein) were incubated with 2 µg of the indicated antibodies in 1 mL immunoprecipitation lysis buffer (NaCl 150 mM, EDTA 5 mM, Tris-HCl 10 mM, Triton X100 0.1%, pH 7.4) for 3 h at 4 °C, followed by incubation with 50 µL of protein A/G agarose beads (sc-2003 Santa Cruz Biotechnology) for 1 h at 4 °C. After low speed centrifugation, $1000 \times g$ for 5 min at room temperature, pellet was washed 3 times in 1 mL immunoprecipitation lysis buffer with low speed centrifugation after each washed. The immunoprecipitates were either assayed for GTPase activity or separated on SDS–PAGE and transferred onto nitrocellulose membranes. The membranes were then probed with the indicated antibodies.

**GTPase activity assay.** Recombinant Drp1 (H00010059-P01), Mfn1 (H00055669-P01), or Mfn2 (H00009927-Q01) from Abnova were incubated with 50 ng of recombinant βIIPKC (ab60841) from Abcam in assay buffer (25 mM Tris-HCl, pH 7.5, 1 mM $CaCl_2$, 20 mM $MgCl_2$, 1 mM DTT, 25 nM ATP) in the presence and absence of phosphatidylserine and diacylglycerol PS/DAG for 30 min at 37 °C. GTPase activity of the proteins was determined using a GTPase assay kit (602–0120, Novus Biologicals) according to manufacturer's instructions. To determine GTPase activity of Mfn1 and Mfn2 in the tissue lysate, a total of 0.25 mg cardiac mitochondria were immunoprecipitated overnight with 2 µg of antibodies against Mfn1 or Mfn2 and 50 µL of Protein A/G–agarose[33]. After three washes with lysis buffer and three washes with GTPase buffer (50 mM Tris [pH 7.5], 2.5 mM $MgCl_2$, and 0.02% 2- mercaptoethanol), the beads were incubated with 0.5 mM GTP at 30 °C for 1 h. The released free phosphate was quantified using the GTPase assay kit, as above.

**Phosphorylation assay.** Twenty-five nanogram of recombinant Mfn1 (H00055669-P01, Abnova) was incubated with 50 ng of recombinant βIPKC (ab60840, Abcam) or βIIPKC (ab60841, Abcam) in assay buffer (25 mM Tris-HCl, pH 7.5, 1 mM $CaCl_2$, 20 mM $MgCl_2$, 1 mM DTT, and 25 nM ATP) in the presence and absence of phosphatidylserine and diacylglycerol PS/DAG for 30 min at 37 °C. Mfn1 phosphorylation was evaluated using radioactive [P$^{32}$] P$^{32}$-ATP incorporation[11].

**Mass spectrometry analysis.** The identification of Mfn1 phosphorylation at βIIPKC site was performed by incubating recombinant Mfn1 with recombinant βIIPKC plus PS/DG/Ca (PKC activators). Phosphopeptide spectra of Mfn1 was examined using LC/MS/MS. LTQ Orbitrap Velos mass spectrometer was used, the nano HPLC was an Eksigent nano 2D with a flow rate of 0.6 µL min$^{-1}$. In solution digestion was done with trypsin overnight. Peptides were further purified on a stagetip using C18 matrix. Data-dependent acquisition was used and the top 10 most intense peptides identified in the survey scan were selected for fragmentation using higher energy collisional-induced dissociation (HCD).

**Proteinase K digestion.** For proteinase K digestion, isolated mitochondria were suspended in isotonic mitochondrial buffer and incubated for 15 min with 5 µg mL$^{-1}$ proteinase K at 37 °C. Digestion was terminated by adding 2 mM phenylmethylsulfonyl fluoride. Mitochondrial proteins were separated on SDS–PAGE and transferred onto nitrocellulose membranes. The membranes were then probed with the indicated antibodies.

**Transmission electron microscopy.** Cardiac samples and culture neonatal rat cardiomyocytes were fixed, embedded, and stained[50]. Small blocks (5 blocks per animal) from a cardiac remote area and from neonatal cardiomyocyte pellets were fixed with 2% glutaraldehyde and 4% paraformaldehyde in sodium cacodylate buffer, pH 7.3 for 1 h at room temperature and cut into ~1 mm$^3$ blocks. After several buffer washes, samples were post-fixed in 2% osmium tetroxide and 1% uranyl acetate for 2 h, rinsed in water, dehydrated through ascending concentrations of ethanol followed by 100% acetone, and then infiltrated and embedded in Eponate 12. Final blocks were used to evaluate intermyofibrillar mitochondrial number and size. Images were acquired using a JEOL1230 Gatan 967 CCD transmission electron microscope at 80 kV and a Gatan Orius 4k×4k digital camera (Gatan). We performed quantitative analysis in at least 10 fields per rat (3 rats per group) and at least 15 cardiomyocytes per group (3 independent experiments) using Image J software. The analysis was done in a blinded fashion.

**Cell culture.** WT (ATCC CRL-2991) and Mfn1 knockout (ATCC CRL-2992) MEFs were acquired from ATCC and cultured in Dulbecco's Modification of Eagle's Medium/Ham's F-12 50/50 Mix supplemented with 10% FBS, 100 U mL$^{-1}$ penicillin, and 100 µg mL$^{-1}$ streptomycin. Site-directed mutagenesis was performed to generate Mfn1 S86A, S284A and S290A variants using a QuickChange site-directed mutagenesis kit (Stratagene). Mutants were sequenced to ensure that no additional mutations were introduced and then transfected into Mfn1 knockout MEFs using Lipofectamine transfection reagent (Thermo). Primary cultures of neonatal cardiac myocytes were prepared from the heart of 1-day-old rats by gentle digestion at 37 °C using a cell isolation kit protocol (ac-7018, Cellutron)[39]. Cells were cultured in the presence of 0.1 mmol L$^{-1}$ bromodeoxyuridine on primary tissue culture dishes (BD Falcon) or on laminin-coated slides in Dulbeco's modified Eagle's medium with 10% fetal bovine serum for 4 days. Primary cultures of adult cardiac myocytes were prepared from the heart of 2-month-old rats by enzymatic digestion with collagenase Type II (Worthington) using a Langendorff system[51]. Cells were cultured in M199 medium supplemented with 2 mg ml$^{-1}$ BSA, 1 µM insulin, 2 mM carnitine hydrochloride, 5 mM creatine, and 5 mM taurine for 2 h. Small interfering siRNA duplexes for βIIPKC or negative control were obtained from Amcon Biotechnology. Cardiac myocytes at 50% confluency were transfected for 48 h with siRNA of βIIPKC or negative control siRNA using Lipofectamine 2000 (Invitrogen), according to the manufacturer's instructions.

**Immunocytochemistry.** Cells cultured on coverslips were washed with cold PBS, fixed in 4% formaldehyde and permeabilized with 0.1% Triton X-100. After incubation with 2% normal goat serum (to block non-specific staining), fixed cells were incubated overnight at 4 °C with antibodies against Tom20 (sc-11021, Santa Cruz Biotechnology, dilution 1:500). Cells were washed with PBS and incubated for 60 min with FITC-labeled goat anti-rabbit antibody (65–6111, Invitrogen, dilution 1:500) followed by incubation with Hoechst dye (1:10000) for 10 min. Coverslips were mounted and slides were imaged using a DeltaVision OMX SR wide field deconvolution fluorescence microscope (Applied Precision Inc). Quantification was carried out using Image J software.

**Measurement of cell viability.** Cell viability was measured using an in vitro toxicology assay, MTT-based kit (Sigma), according to the manufacturer's instructions. Cell death was detected by measuring lactate dehydrogenase (LDH) activity in the cell culture media. For each assay, the media was combined with Sodium pyruvate (30 mM), NADH (6.6 mM), and Tris-HCL (0.2 M, pH 7.3), in a final volume to 200 µl. LDH activity was determined by a decrease in absorbance at 340 nm for 10 min at 25 °C, resulting from the oxidation of NADH.

**Peptide synthesis.** Peptides were synthesized using solid-phase chemistry on a Liberty Microwave Peptide Synthesizer (CEM Corporation, Matthews)[7]. Peptides were synthesized as one polypeptide with TAT$_{47–57}$ carrier in the following order: N-terminus–TAT–spacer (Gly-Gly)–cargo–C-terminus.

**Statistical analysis.** Data are presented as means ± standard error of the mean (SEM). Data normality was assessed through Shapiro-Wilk test. One-way analysis of variance (ANOVA) was used to analyze data presented in Figs. 1–6. Two-way ANOVA for repeated measures was used to analyze data depicted in Fig. 7. Whenever significant $F$-values were obtained, Duncan adjustment was used for multiple comparison purposes. GraphPad Prism Statistics was used for the analysis and statistical significance was considered achieved when the value of $P$ was <0.05.

**Reporting summary**. Further information on experimental design is available in the Nature Research Reporting Summary linked to this article.

## Data availability

The data that support the findings of this study are available from the corresponding author upon reasonable request.

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

## Acknowledgements

This work was supported by the National Institute of Health Grant HL52141 to D.M.-R.; Fundação de Amparo à Pesquisa do Estado de São Paulo—Brasil (FAPESP #2009/12349-2, #2010/00028-4, #2010/51906-1, #2012/05765-2, #2015/20783-5, #2016/01633-5, #2015/22814-5, #2016/09611-0, #2017/16694-2, #2017/11142-1, and #2017/16540-5 to J. C.B.F.), Conselho Nacional de Pesquisa e Desenvolvimento (CNPq)—Brasil [#303281/2015-4, #470880/2012-0, and #407306/2013-7 to J.C.B.F.], Coordenação de Aperfeiçoamento de Pessoal de Nível Superior—Brasil (CAPES)—Finance Code 001; and Instituto Nacional de Ciência e Tecnologia e Centro de Pesquisa e Desenvolvimento de Processos Redox em Biomedicina. We thank Camille C. Caldeira-da-Silva for technical assistance.

## Author contributions

J.C.B.F and D.M.R. designed the study, designed the inhibitor, and wrote the manuscript. J.C.B.F performed functional studies of peptides $\beta II_{V5-3}$ and SAM$\beta$A in vivo and in culture. N.Q. designed SAM$\beta$A peptide and generated structural images. X.Q. performed Co-IP studies and in vitro GTPase assays. J.C.C., V.M.L, B.B.Q., L.H.M.B., and A.J.K performed animal surgeries and structural and functional studies of mitochondria in heart failure. L.R.G.B. performed point mutation studies in MEFs. M.H.D. performed in vitro Co-IP and phosphorylation assays. P.M.M.D. performed echocardiogram measurements.

## Additional information

**Competing interests:** J.C.B.F and D.M.-R. are co-inventors of patent on "Antagonists of mitofusin 1 and beta II PKC association for treating heart failure", PCT/US2019/062854. The remaining authors declare no competing interests.

