## [Peer Review File · Nature Communications]

Reviewers' comments:

Reviewer #1 (Remarks to the Author):

NCOMMS-17-1686146

Previous finding shows an increased β IIPKC activity in failing hearts. However it remains unknown how β IIPKC may lead to heart failure. In the present work, β IIPKC form a complex with mitochondrial fission protein mitofusin 1 in mitochondrial outer membrane. β IIPKC stimulates Serine 86 phosphorylation, mitochondrial fragmentation, mitochondria dysfunction and cardiomyocyte death. P256, a selective peptide inhibitor of β IIPKC-Mfn1 complex, has cardioprotective effects both in vitro and in vivo.

This is novel study with important and interesting results. Some previous work has shown that ERK phosphorylates Treo 562 MFN 1, modulating its participation in apoptosis and mitochondrial fusion (Pyakurel et al. Extracellular regulated kinase phosphorylates mitofusin 1 to control mitochondrial morphology and apoptosis. Mol Cell 58:244-54, 2015. The authors discussed this paper in the discussion.

State of the art. The authors made a completely and updated description of the present status of the role of PKC in heart failure, including their previous work showing that activation of β IIPKC, but not other PKCs, contributes to heart failure pathophysiology. They also described their recent work with the rationally designed peptide against β IIPKC named β IIV5-3. This peptide showed to improve cardiac function in experimental HF models induced by hypertension or MI. This global β IIPKC inhibitor peptide inhibits all β IIPKC activities.

Aim. To discovery downstream β IIPKC substrate involved in heart failure progression

Major points:

1. Incomplete information and/or poor experimental design.

In Figure 1. MI was chosen as model of heart failure. a) Infarct size is not described in this work. b) The control group Sham + β IIV5-3 is missing in Fig 1A-D. c) The authors need to explain with more details how the remote (viable) zone was selected for the biochemical determinations. Did they evaluate markers of cell death (necrosis and/or apoptosis). In electron microscopy studies (Panel E), it is unclear where the evaluation of mitochondrial morphology was done and if subsarcolemmal and/or intermyofibrillar mitochondria were evaluated? Was the response homogenous? On the other hand, mitochondrial dynamics is a concept of redistribution of total mitochondrial volume in small or large units. Normally, this can or cannot change in parallel with/without alterations mitochondrial biogenesis or mitophagy. Both the evaluation of mitochondrial DNA levels and several housekeeping proteins from the inner and outer mitochondrial membranes is required. More important that the evaluation of β IIPKC protein levels is the assessment of its activity.

In Figure 2. The authors use NRVM as model. a) It is unclear the rationale to use Ang II or ceramide in the in vitro studies. B) The study will be strengthened if the authors do some selected experiments in adult mouse cardiomyocytes isolated from mice described in Fig 1 and repeat those described in NRM but using isolated from β IIPKC ko mice. Did you use C2 or C6-ceramide The authors did not use the appropriate controls: DH-C2 or DH-C6 ceramide. The mitochondrial fragmentation in in vitro studies must be also assessed by electron microscopy study.

2. Quality of the data in some graphs is not optimal (i.e. Fig 3C), or some graphs are not completely representative of the images. In same case, a double band for Mfn1 is observed. It's correct? Please provide positive and negative controls for the detection of Mfn1 by western blotting,

3. It is unclear in vivo if this increase in Mfn1 phosphorylation induced by β IIPKC is linked to mitophagy and cardiomyocyte death. This point should be addressed in this work.

4. Translational impact of these results remains unclear.

Minor comments

1. MW markers must be included in all western blots.

2. Scale bars must be added in all corresponding images

Reviewer #2 (Remarks to the Author):

This is a timely and highly significant study that demonstrates a functional link between mitochondrial dysfunction and heart failure. The authors show convincing evidence that this is mediated by the accumulation of β IIPKC on the outer mitochondrial membrane and subsequent phosphorylation of Mfn1 at S86. Armed with this biochemical framework, the authors then rationally designed a small peptide that selectively inhibits the β IIPKC-Mfn1 interaction and demonstrated that treatment with this agent re-established mitochondrial morphology and improved cardiac function in a post-MI model of heart failure. The data presented are logical and comprehensive and the study is highly significant in that it brings together several strands of evidence surrounding progressive heart failure (activation of selective PKC isoforms; mitochondrial fragmentation; depressed mitochondrial respiration) and identifies a single regulatory interaction that can be therapeutically targeted. The work should be of great interest to the community of cardiovascular investigators who are interested in mechanisms of progressive LV dysfunction in models of heart failure. Suggestions for revision follow:

1. The animal model chosen (remodeling post-MI) is well accepted but obviously involves heterogeneous changes in the myocardium (scar/hypertrophy/etc.) Given that the relevant biochemical pathway has been identified and a small molecule has been designed, it might be of interest to make the findings more generalizable by looking at a model of dilated non-ischemic cardiomyopathy. Also, a minor point is that the lack of an effect of P256 on the vasculature should be documented to confirm that changes in blood pressure did not contribute to the change in Efx and EDd.
2. Given that PKC isoforms have been shown to target contractile proteins and to influence cross-bridge cycling, it would be useful to document that changes in myofilament phosphorylation (Tnl, MLC2) are unaffected by the novel interventions proposed.
3. Studies done in neonatal cardiomyocytes (fig 2) are certainly justifiable by virtue of their relative ease of use. However, the authors should rationalize this model (as opposed to adult myocytes) given that several cellular functions linked to metabolism may be different at different developmental stages.

Reviewer #3 (Remarks to the Author):

In the current study, Ferreira et al. explore the role of mitochondrial dynamics, specifically upstream molecular mechanisms affecting the mitochondrial morphology, in cardiac remodeling and heart failure. The experimental data presented support the role of β IIPKC activation and translocation to the outer mitochondrial membrane is pivotal to heart failure progression. Subsequently, the investigators demonstrated that the translocated β IIPKC bound to and phosphorylated the mitochondrial enzyme Mfn1 at S86 located in its GTPase domain. Accordingly, they hypothesize that β IIPKC phosphorylation of Mfn1 is critical for mitochondrial fusion activity causing accumulation of fragmented mitochondria with the release of free radicals, and disturbed the mitochondrial oxygen consumption and oxidative phosphorylation. Furthermore, targeted treatment with P256 peptide, a rationally designed selective inhibitor of the β IIPKC-Mfn1 interaction, prevented mitochondrial fragmentation in cardiac myocytes under stress conditions triggered by angiotensin II. Finally, P256 treatment on the rat model of myocardial infarction-induced heart failure dramatically improved mitochondrial morphology, bioenergetics, redox balance and cardiac function. In contrast, the global β IIPKC inhibitor β IIV5-3 was less effective. The rational design and application of this novel peptide, P256, for direct and specific regulation of protein-protein interaction and heart failure may bear great translational value.

This study was well-designed and carefully executed, providing sufficient evidence to support their hypothesis. The design and application of P256 peptide for the targeted inhibition of β IIPKC-Mfn1 interaction in the setting of heart failure is novel. The effects of P256 in reversing mitochondrial disarray and cardiac function in in-vivo rats of ischemic heart failure have high clinical implications. Finally, there are a few remaining questions with respect to the concept and approach of the study as outlined below.

Comments:

1. Overall throughout the manuscript, the replicates for experiments examining mitochondrial morphology (e.g., Fig. 1e, 2f & 6e) are excellent ($n > 10$). However, the replicates for experiments characterizing mitochondrial function (e.g., Fig. 2a, 2b, & 5h-j) are at a minimum ($n \leq 5$) and a large sample size may be necessary to solidify the results.

2. Treatment with the selective inhibitor of β IIPKC-Mfn1 interaction was commenced at 4 weeks after induction of myocardial infarction. In this setting, P256 was able to preserve mitochondrial dynamics as well as cardiac function, demonstrating its potential to minimize the progress of post-MI heart failure. However, from a clinical standpoint, it may also be interesting to see the in vivo effects of P256 when treatment is commenced immediately after myocardial infarction; would P256 have an effect to prevent cardiac dysfunction post-MI.

3. [Fig. 6j & 6k] The heart EF values of animals the Ctr group before treatment appeared to be lower than those before treatments in the β IIV5-3 and P256 groups, whereas the values of LVEDd in the

Ctr group before treatment appeared to be higher than the before treatments in the β IIV5-3 and P256 groups.

4. The “Title” states “selective inhibition of β IIPKC-mediated mitofusin 1 phosphorylation is sufficient to improve heart failure outcome”; the authors indicate that if β IIPKC-mediated Mfn1 phosphorylation is inhibited, then heart failure outcome will improve. If this title is to remain, then some direct evidence that Mfn1 S86A improves post-MI heart failure remodeling would be required. As P256 inhibits β IIPKC-Mfn1 interaction and improves heart failure outcome, an experiment documenting the inhibition of P256 on Mfn1 phosphorylation would further strengthen the statement. Alternatively, a title emphasizes on targeted inhibition of β IIPKC-Mfn1 interaction would be more accurately to reflect the results.

5. [Abstract], the statement “S86 phosphorylation reduced Mfn1 GTPase activity in vitro” is strong; should be supported by GTPase activity assay on Mfn1 S86A.

6. [Fig. 1 & 2] The presentation of β IIV5-3 should be consistent throughout the manuscript, either with subscript (as in panels of Fig. 1b-g & Fig 2) or without subscript (as in the figure legends and main text).

7. [Fig. 2h] The abbreviation of siRNA negative control (siRNA NC) should be included in the figure legend.

RESPONSE TO THE REVIEWERS

Reviewer #1:

Previous finding shows an increased β IIPKC activity in failing hearts. However it remains unknown how β IIPKC may lead to heart failure. In the present work, β IIPKC form a complex with mitochondrial fission protein mitofusin 1 in mitochondrial outer membrane. β IIPKC stimulates Serine 86 phosphorylation, mitochondrial fragmentation, mitochondria dysfunction and cardiomyocyte death. P256, a selective peptide inhibitor of β IIPKC-Mfn1 complex, has cardioprotective effects both in vitro and in vivo.

This is novel study with important and interesting results. Some previous work has shown that ERK phosphorylates Treo 562 MFN 1, modulating its participation in apoptosis and mitochondrial fusion (Pyakurel et al. Extracellular regulated kinase phosphorylates mitofusin 1 to control mitochondrial morphology and apoptosis. Mol Cell 58:244-54, 2015. The authors discussed this paper in the discussion.

State of the art. The authors made a completely and updated description of the present status of the role of PKC in heart failure, including their previous work showing that activation of β IIPKC, but not other PKCs, contributes to heart failure pathophysiology. They also described their recent work with the rationally designed peptide against β IIPKC named β IIV5-3. This peptide showed to improve cardiac function in experimental HF models induced by hypertension or MI. This global β IIPKC inhibitor peptide inhibits all β IIPKC activities.

Aim. To discovery downstream β IIPKC substrate involved in heart failure progression

Major points:

1. Incomplete information and/or poor experimental design.

In Figure 1. MI was chosen as model of heart failure. a) Infarct size is not described in this work.

R. The infarct size was similar in all animals undergoing permanent coronary artery occlusion (~46%), despite the pharmacological intervention (β IIV5-3, vehicle or P256 peptide, now termed SAM β A). These data (including representative images) are now presented in Supplementary Tables 1 and 3.

b) The control group Sham + β IIV5-3 is missing in Fig 1A-D.

R. We have now included these data in the Supplementary Figure 1. Note that neither cardiac function nor mitochondrial number/size were affected by chronic treatment with peptide β IIV5-3 in the Sham group.

c) The authors need to explain with more details how the remote (viable) zone was selected for the biochemical determinations.

R. We have now included the following text: "Hearts were dissected into infarct and viable zones using the following criteria. The infarct zone was the left ventricular wall that was thinning and included visible infarct scar, and the viable (remote) zone was the part of the left ventricular wall that was reddish-brown and with normal thickness. All biochemical measurements were performed in the left ventricular viable (remote) zone". See page 19 paragraph 1.

Did they evaluate markers of cell death (necrosis and/or apoptosis).

R. We have included a new set of data showing that sustained treatment with either β IIV5-3 or p256 (now SAM β A) peptide was sufficient to reduce cardiac apoptosis (as measured by TUNEL staining) compared to vehicle-treated heart failure animals (see new Supplementary Figure 3C).

In electron microscopy studies (Panel E), it is unclear where the evaluation of mitochondrial morphology was done and if subsarcolemmal and/or intermyofibrillar mitochondria were evaluated? Was the response homogenous?

R. Quantification of mitochondria number and size was done in the intermyofibrillar mitochondria within the viable zone. We have included this information in the manuscript (see page 22 paragraph 1). We also quantified subsarcolemmal and perinuclear mitochondria and the data were similar to intermyofibrillar mitochondria.

On the other hand, mitochondrial dynamics is a concept of redistribution of total mitochondrial volume in small or large units. Normally, this can or cannot change in parallel with/without alterations mitochondrial biogenesis or mitophagy. Both the evaluation of mitochondrial DNA levels and several housekeeping proteins from the inner and outer mitochondrial membranes is required.

R. We have now included a new set of data showing accumulation of outer mitochondrial membrane proteins that was not accompanied by a change in levels of mitochondrial matrix proteins, markers of mitophagy (e.g., p62) or mitochondrial DNA levels in vehicle-treated, but not in β IIV5-3-treated, heart failure animals. See new Supplementary Figure 2.

More important that the evaluation of β IIPKC protein levels is the assessment of its activity.

R. β IIPKC translocation from the cytosol to the particulate active fraction (triton soluble fraction, including mitochondrial membranes) is a reliable and commonly used way to assess PKC activity¹. This method of assessing PKC activation has been validated by our group using cells², rodents^{3,4} and human heart specimens⁴, and reproduced by many other groups^{5,6,7,8,9,10}. Therefore, measurement of β IIPKC translocation to the mitochondrial membrane is a simple and effective way to check its activity.

In Figure 2. The authors use NRVM as model. a) It is unclear the rationale to use Ang II or ceramide in the in vitro studies.

R. We used Ang II and ceramide as different means to induce stress that directly affect mitochondrial fusion-fission and cardiomyocyte viability^{11,12}, and elevated levels of Ang II and ceramide are hallmarks of heart failure in humans^{13,14}. We now provide this justification in the revised manuscript (paragraph 2, page 6).

b) The study will be strengthened if the authors do some selected experiments in adult mouse cardiomyocytes isolated from mice described in Fig 1 and repeat those described in NRM but using isolated from β IIPKC ko mice.

R. We have included a new set of data showing that cardiomyocytes isolated from adult HF animals treated with the β IIV5-3 peptide (according to the protocol described in Figure 1A) have improved contractility compared to vehicle-treated HF animals (see new Figure 1B).

We have also provided new data showing that either knocking down β IIPKC or treatment with β IIV5-3 protected adult cardiomyocytes (isolated from healthy animals) from oxidative stress (a well-known trigger of mitochondrial fragmentation¹⁵; new Figures 2G-H), similar to what we found in neonatal cardiomyocytes (new Figures 2E-F). Finally, we used transmission electron microscopy to validate our findings of mitochondrial fragmentation upon stress using cultured neonatal cardiomyocytes (new Figures 2A-B).

We have not used β IIPKC KO mice because we already used genetic (siRNA) and pharmacological (β IIV5-3 peptide) approaches to validate the contribution of β IIPKC toward mitochondrial fragmentation in both neonatal and adult cardiomyocytes from rats. Including another murine model in this study would not have added value relative to the amount of work that we have already performed and the limitations of using knockout models.

Did you use C2 or C6-ceramide The authors did not use the appropriate controls: DH-C2 or DH-C6 ceramide.

R. We presented only the C2-ceramide data (now fully described in the Figure 2 legend); however, we found that the inactive analogue DH-C2-ceramide had no impact on the levels of Mfn1 and β IIPKC in the mitochondrial fraction.

The mitochondrial fragmentation in in vitro studies must be also assessed by electron microscopy study.

R. We have included quantification of mitochondrial fragmentation in neonatal cardiomyocytes using TEM, as requested (see new Figure 2A).

2. Quality of the data in some graphs is not optimal (i.e. Fig 3C), or some graphs are not completely representative of the images. In same case, a double band for Mfn1 is observed. It's correct? Please provide positive and negative controls for the detection of Mfn1 by western blotting.

R. We removed the Mfn2 WB from the panel (now Figure 3G). Since no changes were observed among the conditions tested, a new image of Mfn2 WB is now provided as new Supplementary Figure 2F. The antibody used for Mfn1 can recognize a double band in tissue samples, as previously described by others^{16, 17}. We also provide here a panel (also new Supplementary Figure 4B) showing positive and negative controls for the commercially available antibodies, using for the detection of Mfn1 by WB, as requested.

3. It is unclear in vivo if this increase in Mfn1 phosphorylation induced by β IIPKC is linked to mitophagy and cardiomyocyte death. This point should be addressed in this work.

R: Elevated Mfn1 phosphorylation induced by β IIPKC was linked to increased apoptosis (see new Supplementary Figure 3C, measured by TUNEL assay), but not mitophagy (see new Supplementary Figure 2D, measured by accumulation of mitophagy markers such as optineurin, NDP52 and p62) in failing hearts. Moreover, global inhibition of β IIPKC (β IIV5-3 peptide) was sufficient to reduce cardiac apoptosis as compared with vehicle-treated heart failure animals (see new Supplementary Figure 3C).

4. Translational impact of these results remains unclear.

R: Considering that human failing hearts present elevated β IIPKC activity^{4, 18} as well as increased levels of Mfn1¹⁹, our work suggests that correcting impaired mitochondrial dynamics by selectively inhibiting β IIPKC-Mfn1 interaction with drugs, such as p256 peptide (now SAM β A), may provide useful treatments for patients with established heart failure.

Minor comments

1. MW markers must be included in all western blots.

R: Done.

2. Scale bars must be added in all corresponding images.

R: Done.

Reviewer #2 (Remarks to the Author):

This is a timely and highly significant study that demonstrates a functional link between mitochondrial dysfunction and heart failure. The authors show convincing evidence that this is mediated by the accumulation of β IIPKC on the outer mitochondrial membrane and subsequent phosphorylation of Mfn1 at S86. Armed with this biochemical framework, the authors then rationally designed a small peptide that selectively inhibits the β IIPKC-Mfn1 interaction and demonstrated that treatment with this agent re-established mitochondrial morphology and improved cardiac function in a post-MI model of heart failure. The data presented are logical and comprehensive and the study is highly significant in that it brings together several strands of evidence surrounding progressive heart failure (activation of selective PKC isoforms; mitochondrial fragmentation; depressed mitochondrial respiration) and identifies a single regulatory interaction that can be therapeutically targeted. The work should be of great interest to the community of cardiovascular investigators who are interested in mechanisms of progressive LV dysfunction in models of heart failure. Suggestions for revision follow:

1. The animal model chosen (remodeling post-MI) is well accepted but obviously involves heterogeneous changes in the myocardium (scar/hypertrophy/etc.) Given that the relevant biochemical pathway has been identified and a small molecule has been designed, it might be of interest to make the findings more generalizable by looking at a model of dilated non-ischemic cardiomyopathy.

R: We appreciate the strength of using another animal model of HF. However, we believe that this is outside the immediate scope of this one. We have included this as a limitation in the discussion.

Also, a minor point is that the lack of an effect of P256 on the vasculature should be documented to confirm that changes in blood pressure did not contribute to the change in Efx and EDd.

R: We have included new data demonstrating that sustained treatment with the p256 peptide (now SAM β A) has no impact on blood pressure in heart failure animals (see new Supplementary Table 3).

2. Given that PKC isoforms have been shown to target contractile proteins and to influence cross-bridge cycling, it would be useful to document that changes in myofilament phosphorylation (Tnl, MLC2) are unaffected by the novel interventions proposed.

R: Our new findings demonstrate that the p256 peptide (now SAM β A) has no effect on cardiac β IIPKC-troponin I interaction and troponin I phosphorylation in heart failure animals (see new Figure 6J).

3. Studies done in neonatal cardiomyocytes (fig 2) are certainly justifiable by virtue of their relative ease of use. However, the authors should rationalize this model (as opposed to adult myocytes) given that several cellular functions linked to metabolism may be different at different developmental stages.

R: We agree and have repeated some experiments using adult rat cardiomyocytes. Note that, both neonatal and adult rat cardiomyocytes responded similarly to genetic (siRNA) or pharmacological (β IIV5-3 peptide) β IIPKC inhibition upon stress. These findings using cardiac cells at different levels of development confirm the cardiac role of β IIPKC in mitochondrial fission-fusion balance.

Reviewer #3 (Remarks to the Author):

In the current study, Ferreira et al. explore the role of mitochondrial dynamics, specifically upstream molecular mechanisms affecting the mitochondrial morphology, in cardiac remodeling and heart failure. The experimental data presented support the role of β IIPKC activation and translocation to the outer mitochondrial membrane is pivotal to heart failure progression. Subsequently, the investigators demonstrated that the translocated β IIPKC bound to and phosphorylated the mitochondrial enzyme Mfn1 at S86 located in its GTPase domain. Accordingly, they hypothesize that β IIPKC phosphorylation of Mfn1 is critical for mitochondrial fusion activity causing accumulation of fragmented mitochondria with the release of free radicals, and disturbed the mitochondrial oxygen consumption and oxidative phosphorylation. Furthermore, targeted treatment with P256 peptide, a rationally designed selective inhibitor of the β IIPKC-Mfn1 interaction, prevented mitochondrial fragmentation in cardiac myocytes under stress conditions triggered by angiotensin II. Finally, P256 treatment on the rat model of myocardial infarction-induced heart failure dramatically improved mitochondrial morphology, bioenergetics, redox balance and cardiac function. In contrast, the global β IIPKC inhibitor β IIV5-3 was less effective. The rational design and application of this novel peptide, P256, for direct and specific regulation of protein-protein interaction and heart failure may bear great translational value.

This study was well-designed and carefully executed, providing sufficient evidence to support their hypothesis. The design and application of P256 peptide for the targeted inhibition of β IIPKC-Mfn1 interaction in the setting of heart failure is novel. The effects of P256 in reversing mitochondrial disarray and cardiac function in in-vivo rats of ischemic heart failure have high clinical implications. Finally, there are a few remaining questions with respect to the concept and approach of the study as outlined below.

Comments:

1. Overall throughout the manuscript, the replicates for experiments examining mitochondrial morphology (e.g., Fig. 1e, 2f & 6e) are excellent (n>10). However, the replicates for experiments characterizing mitochondrial function (e.g., Fig. 2a, 2b, & 5h-j) are at a minimum (n≤5) and a large sample size may be necessary to solidify the results.

R: We have increased the sample size for experiments characterizing mitochondrial function (now n=10 per condition). See new Figures 3A-D.

2. Treatment with the selective inhibitor of β IIPKC-Mfn1 interaction was commenced at 4 weeks after induction of myocardial infarction. In this setting, P256 was able to preserve mitochondrial dynamics as well as cardiac function, demonstrating its potential to minimize the progress of post-MI heart failure. However, from a clinical standpoint, it may also be interesting to see the in vivo effects of P256 when treatment is commenced immediately after myocardial infarction; would P256 has an effect to prevent cardiac dysfunction post-MI.

R: We agree with the reviewer that this is an interesting study, but believe that it is outside the immediate scope of this study (now brought up in the discussion; page 17 paragraph 3). We plan to carry out this study in the future, considering its clinical relevance, as pointed out by reviewer.

3. [Fig. 6j & 6k] The heart EF values of animals the Ctr group before treatment appeared to be lower than those before treatments in the β IIV5-3 and P256 groups, whereas the values of LVEDd in the Ctr group before treatment appeared to be higher than the before treatments in the β IIV5-3 and P256 groups.

R: There is no difference in EF and LVEDd values when comparing control, β IIV5-3 and p256 (now SAM β A) groups before treatment (see figure on the right; data obtained from Figure 7A-B, before treatment).

4. The “Title” states “selective inhibition of β IIPKC-mediated mitofusin 1 phosphorylation is sufficient to improve heart failure outcome”; the authors indicate that if β IIPKC-mediated Mfn1 phosphorylation is inhibited, then heart failure outcome will improve. If this title is to remain, then some direct evidence that Mfn1 S86A improves post-MI heart failure remodeling would be required. As P256 inhibits β IIPKC-Mfn1 interaction and improves heart failure outcome, an experiment documenting the inhibition of P256 on Mfn1 phosphorylation would further strengthen the statement. Alternatively, a title emphasizes on targeted inhibition of β IIPKC-Mfn1 interaction would be more accurately to reflect the results.

R: We agree with the reviewer and have changed the title accordingly. We also provided new evidence that p256 (now SAM β A) reduces cardiac β IIPKC-Mfn1 interaction and Mfn1 phosphorylation in heart failure rats (see new Figure 6J).

5. [Abstract], the statement “S86 phosphorylation reduced Mfn1 GTPase activity in vitro” is strong; should be supported by GTPase activity assay on Mfn1 S86A.

R: We agree and have removed this sentence from the abstract as the study focused more on the β IIPKC-Mfn1 interaction.

6. [Fig. 1 & 2] The presentation of β IV5-3 should be consistent throughout the manuscript, either with subscript (as in panels of Fig. 1b-g & Fig 2) or without subscript (as in the figure legends and main text).

R: Done.

7. [Fig. 2h] The abbreviation of siRNA negative control (siRNA NC) should be included in the figure legend.

R: Done.

References

1. Ron D, Mochly-Rosen D. An autoregulatory region in protein kinase C: the pseudoanchoring site. *Proc Natl Acad Sci U S A* **92**, 492-496 (1995).
2. Disatnik MH, Boutet SC, Lee CH, Mochly-Rosen D, Rando TA. Sequential activation of individual PKC isozymes in integrin-mediated muscle cell spreading: a role for MARCKS in an integrin signaling pathway. *J Cell Sci* **115**, 2151-2163 (2002).
3. Palaniyandi SS, Ferreira JCB, Brum PC, Mochly-Rosen D. PKCbetaII inhibition attenuates myocardial infarction induced heart failure and is associated with a reduction of fibrosis and pro-inflammatory responses. *Journal of cellular and molecular medicine* **15**, 1769-1777 (2011).
4. Ferreira JC, Boer BN, Grinberg M, Brum PC, Mochly-Rosen D. Protein quality control disruption by PKCbetaII in heart failure; rescue by the selective PKCbetaII inhibitor, betaIIV5-3. *PLoS One* **7**, e33175 (2012).
5. Hocevar BA, Fields AP. Selective translocation of beta II-protein kinase C to the nucleus of human promyelocytic (HL60) leukemia cells. *J Biol Chem* **266**, 28-33 (1991).
6. Wang HY, Pisano MR, Friedman E. Attenuated protein kinase C activity and translocation in Alzheimer's disease brain. *Neurobiol Aging* **15**, 293-298 (1994).
7. Feng X, Becker KP, Stribling SD, Peters KG, Hannun YA. Regulation of receptor-mediated protein kinase C membrane trafficking by autophosphorylation. *J Biol Chem* **275**, 17024-17034 (2000).
8. Majumder PK, *et al.* Mitochondrial translocation of protein kinase C delta in phorbol ester-induced cytochrome c release and apoptosis. *J Biol Chem* **275**, 21793-21796 (2000).
9. Uecker M, Da Silva R, Grampp T, Pasch T, Schaub MC, Zaugg M. Translocation of protein kinase C isoforms to subcellular targets in ischemic and anesthetic preconditioning. *Anesthesiology* **99**, 138-147 (2003).
10. Caruso M, *et al.* Activation and mitochondrial translocation of protein kinase Cdelta are necessary for insulin stimulation of pyruvate

- dehydrogenase complex activity in muscle and liver cells. *J Biol Chem* **276**, 45088-45097 (2001).
11. Parra V, *et al.* Changes in mitochondrial dynamics during ceramide-induced cardiomyocyte early apoptosis. *Cardiovascular research* **77**, 387-397 (2008).
 12. Qi J, *et al.* Mitochondrial Fission Is Required for Angiotensin II-Induced Cardiomyocyte Apoptosis Mediated by a Sirt1-p53 Signaling Pathway. *Front Pharmacol* **9**, 176 (2018).
 13. Sernerri GG, *et al.* Cardiac angiotensin II formation in the clinical course of heart failure and its relationship with left ventricular function. *Circ Res* **88**, 961-968 (2001).
 14. Peterson LR, *et al.* Ceramide Remodeling and Risk of Cardiovascular Events and Mortality. *J Am Heart Assoc* **7**, (2018).
 15. Wu S, Zhou F, Zhang Z, Xing D. Mitochondrial oxidative stress causes mitochondrial fragmentation via differential modulation of mitochondrial fission-fusion proteins. *FEBS J* **278**, 941-954 (2011).
 16. Morciano G, *et al.* Mcl-1 involvement in mitochondrial dynamics is associated with apoptotic cell death. *Mol Biol Cell* **27**, 20-34 (2016).
 17. Wang J, Chen H, Liu Y, Zhou W, Sun R, Xia M. Retinol binding protein 4 induces mitochondrial dysfunction and vascular oxidative damage. *Atherosclerosis* **240**, 335-344 (2015).
 18. Bowling N, *et al.* Increased protein kinase C activity and expression of Ca²⁺-sensitive isoforms in the failing human heart. *Circulation* **99**, 384-391 (1999).
 19. Chen L, Gong Q, Stice JP, Knowlton AA. Mitochondrial OPA1, apoptosis, and heart failure. *Cardiovascular research* **84**, 91-99 (2009).

REVIEWERS' COMMENTS:

Reviewer #1 (Remarks to the Author):

Although it is known that β IIPKC activity is increased in failing hearts, it remains unclear how this kinase may lead to heart failure. The present revised MS shows that β IIPKC phosphorylates and forms a complex with mitofusin 1 in mitochondrial outer membrane, stimulating mitochondrial fragmentation and dysfunction and cardiomyocyte death. The MS also showed that a rationally designed SAM β A, a peptide that selectively antagonizes Mfn1- β IIPKC association, has cardioprotective effects both in vitro and in vivo.

These were my previous four major criticisms:

1. Incomplete information and/or poor experimental design.
2. Quality of the data in some graphs is not optimal-
3. It is unclear in vivo if this increase in Mfn1 phosphorylation induced by β IIPKC is linked to mitophagy and cardiomyocyte death.
4. Translational impact of these results remains unclear.

I read the revised MS and response letter and the authors made an excellent job- They sincerely responded in a very complete and thoughtful manner to all of my comments, suggestions, concerns and criticisms. I do not have further comments.

Reviewer #2 (Remarks to the Author):

No residual concerns

Reviewer #3 (Remarks to the Author):

The authors have addressed most of my concerns. The only remaining point is the title: Since the key evidence supporting the claim came from experiments using the rat model, this should be added to the title. The suggested title, which reflects accurately the findings, should be "...SAMBA, improves heart failure outcome in a rat model of MI".